# Denitrification as the predominant process in nitrous oxide production in the water column of two eutrophic reservoirs

Elizabeth Leon-Palmero<sup>1,2a</sup>, Claudia Frey<sup>3</sup>, Bess B. Ward<sup>2</sup>, Rafael Morales-Baquero<sup>1</sup>, and Isabel Reche<sup>1,4</sup>

- <sup>1</sup>Departamento de Ecología and Instituto del Agua, Universidad de Granada, Granada, E-18071, Spain
- <sup>2</sup>Department of Geosciences, Princeton University, Princeton, NJ, E-08544, USA
- <sup>3</sup>Department of Environmental Science, University of Basel, Basel, E-4056, Switzerland
- <sup>4</sup>Research Unit Modeling Nature (MNat), Universidad de Granada, Granada, E-18071, Spain
- <sup>a</sup>Current address: Department of Geosciences, Princeton University, Princeton, NJ, E-08544, USA
- 10 Correspondence to: Elizabeth Leon-Palmero (el23@princeton.edu)

**Abstract.** Reservoirs are important sites for nitrogen cycling and a significant global source of the potent greenhouse gas nitrous oxide (N<sub>2</sub>O) to the atmosphere. They receive nitrogen inputs from agriculture and urban sources, boosting the production of N<sub>2</sub>O by nitrification, denitrification, and photochemodenitrification. However, existing estimates of N<sub>2</sub>O production in reservoirs are uncertain because previous studies have mainly focused on N<sub>2</sub>O in rivers or lake sediments, often overlooking the water column of lentic systems. Here, we employed stable isotope tracer incubations alongside analyses of in situ natural abundance of nitrogen pools and functional genes involved in nitrification (*amoA*) and denitrification (*nirS*), to study N<sub>2</sub>O production in the water column of two eutrophic reservoirs with contrasting morphometries. We used <sup>15</sup>N-NH<sub>4</sub>+ and <sup>15</sup>N-NO<sub>3</sub>- tracers to quantify rates of N<sub>2</sub>O production, nitrification, and nitrate reduction at the beginning and the end of the stratification period. Notably, nitrate concentration decreased by up to 49% over the two months. N<sub>2</sub>O production from ammonium ranged from 0.02 to 48.6 nmol-N L<sup>-1</sup> d<sup>-1</sup>, while N<sub>2</sub>O from nitrate varied from 0.2 to 61.0 nmol-N L<sup>-1</sup> d<sup>-1</sup>. High rates of nitrification, nitrate reduction to nitrite, and rapid nitrite turnover were observed, with total N<sub>2</sub>O production significantly correlated with the abundance of the *nirS* gene. A strong positive correlation was found between δ<sup>15</sup>N-NO<sub>2</sub>- and both N<sub>2</sub>O concentration and *nirS* abundance. Overall, these findings suggest that reservoirs are active sites for N<sub>2</sub>O production and N loss, with denitrification playing a significant role in the water column.

## 25 1 Introduction

Reservoirs created by damming rivers are an important global source of the greenhouse gas nitrous oxide (N<sub>2</sub>O) to the atmosphere (Li et al., 2024; Wang et al., 2023). N<sub>2</sub>O is about 273 times as potent as carbon dioxide for atmospheric warming on a 100-year time horizon (IPCC, 2021), and is the main driver of stratospheric ozone depletion (Ravishankara et al., 2009). Reservoirs receive substantial nitrogen (N) loading from agriculture and urban areas in their watersheds, processing it throughout different microbial and abiotic pathways, and then emitting back a fraction to the atmosphere as dinitrogen gas (N<sub>2</sub>) and, significantly, N<sub>2</sub>O (Leon-Palmero et al., 2025; León-Palmero, 2023). Reservoirs accounted for 50% (i.e., 0.44 Tg N

yr<sup>-1</sup>) of the total increase in N<sub>2</sub>O emissions from inland waters between 1900 and 2010 (i.e., 0.89 Tg N yr<sup>-1</sup>) (Wang et al., 2023). This rapid rise in N<sub>2</sub>O emissions from reservoirs is linked to the growing number of reservoirs worldwide (Lehner et al., 2011), as well as an increase in N<sub>2</sub>O production within these reservoirs (Wang et al., 2023). Nevertheless, the existing emission estimates are still uncertain because they are based on limited datasets. Reservoirs have not been studied as extensively as other inland waters, such as lakes or rivers, even though they process a disproportionately high fraction of the N compared to other aquatic systems (Harrison et al., 2009), leading to high N<sub>2</sub>O production rates and subsequent emissions (Beaulieu et al., 2015; León-Palmero et al., 2020a, 2023; Rodríguez-Velasco et al., 2024). Therefore, it is crucial to understand the factors controlling N<sub>2</sub>O production in reservoirs, especially considering the global increase in reservoir construction (Zarfl et al., 2015).

Microbial transformations that lead to the production and consumption of N<sub>2</sub>O include ammonia oxidation, nitrifier denitrification, and denitrification, and they are all affected by the availability of N-substrates, oxygen concentration, and phosphorus availability (Beaulieu et al., 2015; Codispoti, 2010; Ji et al., 2018; León-Palmero et al., 2023). N<sub>2</sub>O is a byproduct of ammonia oxidation to nitrite (i.e., first step of nitrification), which is performed by ammonia-oxidizing bacteria (AOB) and ammonia-oxidizing archaea (AOA) in oxygenated waters (Könneke et al., 2005; Kowalchuk and Stephen, 2001), with the latter dominating in Mediterranean reservoirs (León-Palmero et al., 2023). At low oxygen concentrations, nitrifiers increase the yield of N<sub>2</sub>O production, relative to the ammonium (NH<sub>4</sub><sup>+</sup>) oxidized, by nitrifier denitrification (via AOB), hybrid formation (AOA), or hydroxylamine oxidation (AOA), although some details of the reactions remain unresolved (Stein, 2019; Wan et al., 2023; Ward, 2013). Lastly, denitrification is the reduction of nitrate (NO<sub>3</sub><sup>-</sup>) to nitrite (NO<sub>2</sub><sup>-</sup>), nitric oxide (NO), N<sub>2</sub>O, and N<sub>2</sub>, coupled to organic matter oxidation. Hence, denitrification can act as a source or sink of N<sub>2</sub>O depending on the rate of N<sub>2</sub>O reduction to N<sub>2</sub>, which is catalyzed by the enzyme N<sub>2</sub>O reductase. Denitrification is an anaerobic pathway, and oxygen regulates the activity of the denitrifying enzymes, especially the N<sub>2</sub>O reductase (Bonin et al., 1989; Zumft, 1997). However, many bacteria can denitrify in both oxic and anoxic conditions (Hochstein et al., 1984; Lloyd et al., 1987), and the presence of denitrifying bacteria has been demonstrated in the oxic and anoxic water column of lakes (Junier et al., 2008; Kim et al., 2011; Pajares et al., 2017) and reservoirs (León-Palmero et al., 2023).

Moreover, other specific factors may influence the production, accumulation, and emission of N<sub>2</sub>O in reservoirs, such as morphometry (i.e., depth and shape) and water residence time (Hayes et al., 2017; Liang et al., 2019). The morphometry of a reservoir and water residence time affect thermal and oxygen stratification, as well as N<sub>2</sub>O storage in the water column. Deep reservoirs can produce and accumulate large concentrations of N<sub>2</sub>O in the hypolimnion during thermal stratification, particularly under anoxic conditions and high N concentrations. In contrast, denitrification can be a sink of N<sub>2</sub>O in the anoxic hypolimnion when N concentration is low (Beaulieu et al., 2015; León-Palmero et al., 2023). Shallow systems tend to emit N<sub>2</sub>O continuously due to weak thermal stratification and less capacity to accumulate N<sub>2</sub>O. Further studies on N<sub>2</sub>O production in the water column of reservoirs with different morphometries are required to improve our knowledge of N<sub>2</sub>O emissions.

In this study, we combined stable isotope tracer incubations with analyses of the *in situ* natural abundances of the N pools and functional genes involved in  $N_2O$  cycling to quantify  $N_2O$  production rates and trace the origin of the  $N_2O$  in the water column

70

of two contrasting reservoirs. We used <sup>15</sup>N-NH<sub>4</sub><sup>+</sup> to quantify the rates of N<sub>2</sub>O production from NH<sub>4</sub><sup>+</sup>, and ammonia oxidation to nitrite and nitrate; and <sup>15</sup>N-NO<sub>3</sub><sup>-</sup> to trace the formation of N<sub>2</sub>O and NO<sub>2</sub><sup>-</sup> from NO<sub>3</sub><sup>-</sup> reduction. We performed the incubations at three depths at the beginning and at the end of the summer stratification. We selected a shallow and a deep reservoir (Cubillas and Iznájar, respectively) located in watersheds with high N inputs but contrasting morphometries, both of them monomictic with significant emissions and concentrations of N<sub>2</sub>O (León-Palmero et al. 2020a, 2023).

#### 2 Material and Methods

## 2.1 Study reservoirs, morphometry, and watersheds

This study was conducted in southeastern Spain (Fig. S1) within two monomictic reservoirs with contrasting morphometries. Cubillas (37.27°N, 3.68°W) is a small and shallow reservoir with a surface area of 1.94 km² and a total capacity of 19 hm³ (mean depth = 9.66 m). Iznájar (37.26°N, 4.33°W) is a big and deep reservoir with a surface area of 26 km² and a total capacity of 981 hm³ (mean depth = 37.55 m) (open database IDEAndalucia; <a href="http://www.ideandalucia.es/portal/web/ideandalucia/">http://www.ideandalucia.es/portal/web/ideandalucia/</a>). Both reservoirs are impacted by large areas of agriculture and urban areas in their watersheds, which results in large inputs of N and phosphorus (León-Palmero et al. 2020a, 2023). More information about the watersheds, morphometry, and water column characterization is provided in previous studies (e.g., León-Palmero et al., 2020a, b).

We sampled the water column of these reservoirs at the beginning (July 4<sup>th</sup> and 9<sup>th</sup>) and the end (September 5<sup>th</sup> and 7<sup>th</sup>) of the summer stratification in 2018. During the study period, intense human usage caused a decline in the volume and water level in both reservoirs, although this decline was more evident in the smaller reservoir (i.e., Cubillas). Cubillas reservoir decreased in volume from 17 hm<sup>3</sup> in July to 11 hm<sup>3</sup> in September and experienced a 3.4 m reduction in the water level. The hydraulic residence time during the study period was 83 days. Iznájar reservoir decreased in volume from 575 hm<sup>3</sup> in July to 480 hm<sup>3</sup> in September, with a 5.4 m reduction in the water level. The hydraulic residence time was 255 days during this period. The reservoir volumes and water levels on specific dates were obtained from the Confederación Hidrográfica del Guadalquivir open database (CHG; https://www.chguadalquivir.es/saih/).

## 2.2 Vertical profiles and Biogeochemical characterization

Using a Sea-Bird 19plus CTD profiler, we obtained continuous measurements of temperature (°C), dissolved oxygen (DO, μmol L<sup>-1</sup>), and conductivity (μS cm<sup>-1</sup>) in the reservoirs' open waters. We then sampled three depths (epilimnion, oxycline, and hypolimnion or bottom waters) with a 5-L UWITEC bottle for further analyses and incubation experiments.

Samples for dissolved N<sub>2</sub>O analysis were taken in 250-mL air-tight Winkler bottles in duplicate, preserved with a solution of HgCl<sub>2</sub> (final concentration 1 mmol L<sup>-1</sup>) to inhibit biological activity, and sealed with Apiezon® grease to prevent gas exchange. Samples were stored in the dark at a controlled temperature (25 °C) until analysis. Dissolved N<sub>2</sub>O concentration was measured using headspace equilibration in a 50-mL air-tight glass syringe in triplicate in each bottle from each sample. N<sub>2</sub>O concentration




was quantified using a daily calibrated gas chromatograph (Bruker® GC-450) as detailed in a previous study (León-Palmero et al., 2023).

Water samples for chemical and biological analysis were maintained at 4 °C until arrival at the laboratory. The particulate material from 500 to 1000 mL of water was filtered through pre-combusted 0.7 μm pore-size Whatman GF/F glass-fiber filters. Chlorophyll *a* (Chl *a*) was extracted from the filtered material and measured following the standard method (APHA 1992). To obtain the cumulative Chl *a* (a proxy for fresh organic matter exported to the water column) in the whole water column (mg Chl *a* m<sup>-2</sup>), from the discrete depths, we summed the concentration of Chl *a* of each stratum using the trapezoidal rule (León-Palmero et al. 2020b). Dissolved organic carbon (DOC), NO<sub>3</sub>-, NO<sub>2</sub>-, and ammonium (NH<sub>4</sub>+) were assayed in the filtered water. Samples for DOC determination were acidified with phosphoric acid (final pH < 2) and measured by high–temperature catalytic oxidation using a Shimadzu total organic carbon analyzer (Model TOC-V CSH) (Álvarez-Salgado and Miller, 1998). NO<sub>3</sub>- concentration was assayed using the UV spectrophotometric method at the wavelength of 220 nm and correcting for DOC absorbance at 275 nm (APHA 1992). NO<sub>2</sub>- and NH<sub>4</sub>+ concentrations were measured by Inductively Coupled Plasma Optical Emission Spectrometry at the Centro de Instrumentración Científica of the Universidad de Granada.

#### 2.3 Functional genes

The abundance of unique functional genes involved in N<sub>2</sub>O cycling was quantified using quantitative PCR (qPCR), similarly to a previous study (León-Palmero et al., 2023). DNA was extracted according to Boström et al. (2004), and used in PCR to determine presence, and in qPCR to assess gene abundance. We used standard reaction mix recipes, thermocycling conditions, and primer requirements specified by the manufacturer. Specific primers were selected from studies performed in natural freshwater samples when available. DNA from pure cultures was used as positive controls and for qPCR standard preparation.
We targeted ammonia oxidizers using the archaeal *amoA* gene, as AOA dominated over AOB in these reservoirs (León-Palmero et al., 2023). Comammox *amoA* genes were targeted in PCR assays using degenerate PCR primers for clades A and B (Pjevac et al., 2017), but no positive control could be used in this case. The *nirS* gene abundance was used as a proxy for denitrifiers, while *nosZ* gene abundance, assessed only at the deepest layer, addressed only bacteria reducing N<sub>2</sub>O to N<sub>2</sub>. More details on the qPCR quantification, primers, specific conditions, standards, and positive controls are provided in the
Supplementary Material.

## 2.4 Experimental setup of <sup>15</sup>N tracer incubations

Reservoir water from the three depths was drawn from the sampling bottle into 60-mL glass serum bottles after overflow. Once in the lab, samples from oxic water depths (refer to Table 1) were purged uncapped for 2 min to remove excess N<sub>2</sub>O, and a headspace with ambient air was maintained after being exposed to ambient air for 30 min. Samples from anoxic waters were sealed with butyl rubber septa and crimped with aluminum seals immediately after filling. In these samples, a 3-mL helium headspace was retained after purging for 4 min. The serum bottles were weighed before and after filling them to account for the exact water volume in each sample. Table 1 compiles the incubation setup, conditions, and concentration of inorganic




nitrogen added in each treatment. In the first treatment, we injected nine bottles from the same depth with  $^{15}\text{N-NH}_4^+$  tracer ( $^{15}\text{NH}_4\text{Cl} \ge 98$  atom % 15N, Sigma Aldrich) to a final concentration of 0.5  $\mu$ mol L<sup>-1</sup>, obtaining a fraction labeled of the substrate pools between 0.1 and 1.0. In this treatment, we also added  $^{14}\text{N-NO}_3^-$ , equivalent to 0.10 of the NO<sub>3</sub><sup>-</sup> pool. In the second treatment,  $^{15}\text{N-NO}_3^-$  tracer (K<sup>15</sup>NO<sub>3</sub>, 98 atom % 15N, Sigma Aldrich) was injected to obtain a fraction labeled of the NO<sub>3</sub><sup>-</sup> pool about 0.10. We also added  $^{14}\text{N-NH}_4^+$  to a final concentration of 0.5  $\mu$ mol L<sup>-1</sup>. Samples were incubated in the dark at the *in situ* temperatures from 13 to 26 °C (Table 1).

The first treatment ( $^{15}$ N-NH<sub>4</sub><sup>+</sup> +  $^{14}$ N-NO<sub>3</sub><sup>-</sup>) was performed at all the depths (n=12), but the second treatment ( $^{15}$ N-NO<sub>3</sub><sup>-</sup> +  $^{14}$ N-NH<sub>4</sub><sup>+</sup>) was performed only at the oxycline and hypolimnion (n=7, Table 1). Incubations were terminated by adding 0.1 mL saturated mercuric chloride (HgCl<sub>2</sub>) to two bottles at t<sub>0</sub> ( $\approx$  0.25 h), two at t<sub>1</sub> ( $\approx$  2-3 h), two at t<sub>2</sub> ( $\approx$  12 h), and three at t<sub>3</sub> ( $\approx$  24 h). All samples were stored at room temperature in the dark and shipped to the laboratory at Princeton University for further analysis.

**Table 1.** Incubation conditions and concentration of inorganic nitrogen compounds added in each treatment. Concentrations are measured in  $\mu$ mol-N L<sup>-1</sup>. np=not performed. More details are provided in the main text.

| Reservoir               | #ID | Depth              | Incubation<br>temp. (°C) | Oxygen<br>conditions |                                                                       | ment 1<br>-12)                                                        | Treatment 2 (n=7)                                                     |                                                                       |
|-------------------------|-----|--------------------|--------------------------|----------------------|-----------------------------------------------------------------------|-----------------------------------------------------------------------|-----------------------------------------------------------------------|-----------------------------------------------------------------------|
| Reservoir               | #ID |                    |                          |                      | <sup>15</sup> NH <sub>4</sub> <sup>+</sup><br>(μmol L <sup>-1</sup> ) | <sup>14</sup> NO <sub>3</sub> <sup>-</sup><br>(μmol L <sup>-1</sup> ) | <sup>14</sup> NH <sub>4</sub> <sup>+</sup><br>(μmol L <sup>-1</sup> ) | <sup>15</sup> NO <sub>3</sub> <sup>-</sup><br>(μmol L <sup>-1</sup> ) |
| 0.131                   | #1  | Epilimnion (2 m)   | $25 \pm 0.5$             | Oxic                 | 0.5                                                                   | 35.0                                                                  | np                                                                    |                                                                       |
| Cubillas<br>(July)      | #2  | Oxycline (7 m)     | $20\pm0.5$               | Oxic                 | 0.5                                                                   | 30.0                                                                  | 0.5                                                                   | 30.0                                                                  |
|                         | #3  | Bottom (9.5 m)     | $18 \pm 0.5$             | Anoxic               | 0.5                                                                   | 25.0                                                                  | 0.5                                                                   | 25.0                                                                  |
| Cubillas<br>(September) | #4  | Epilimnion (0.5 m) | $24 \pm 0.5$             | Oxic                 | 0.5                                                                   | 18.0                                                                  | np                                                                    |                                                                       |
|                         | #5  | Epilimnion (2.5 m) | $24 \pm 0.5$             | Oxic                 | 0.5                                                                   | 17.0                                                                  | np                                                                    |                                                                       |
|                         | #6  | Bottom (6.2 m)     | $24 \pm 0.5$             | Anoxic               | 0.5                                                                   | 13.0                                                                  | 0.5                                                                   | 13.0                                                                  |
| Iznájar<br>(July)       | #7  | Epilimnion (3 m)   | $26\pm0.5$               | Oxic                 | 0.5                                                                   | 35.0                                                                  | np                                                                    |                                                                       |
|                         | #8  | Oxycline (8 m)     | $22\pm0.5$               | Oxic                 | 0.5                                                                   | 35.0                                                                  | 0.5                                                                   | 35.0                                                                  |
|                         | #9  | Hypolimnion (20 m) | $13 \pm 0.5$             | Anoxic               | 0.5                                                                   | 35.0                                                                  | 0.5                                                                   | 35.0                                                                  |
| Iznájar<br>(September)  | #10 | Epilimnion (5 m)   | $26\pm0.5$               | Oxic                 | 0.5                                                                   | 33.0                                                                  | np                                                                    |                                                                       |
|                         | #11 | Oxycline (11 m)    | $26 \pm 0.5$             | Anoxic               | 0.5                                                                   | 31.0                                                                  | 0.5                                                                   | 31.0                                                                  |
|                         | #12 | Hypolimnion (23 m) | $15\pm0.5$               | Anoxic               | 0.5                                                                   | 34.0                                                                  | 0.5                                                                   | 34.0                                                                  |

## 2.5 <sup>15</sup>N-N<sub>2</sub>O production rates

The total N<sub>2</sub>O in each incubation bottle was extracted by purging with helium for 35 min at 38 mL min<sup>-1</sup>. Then, N<sub>2</sub>O was trapped by liquid nitrogen and isolated from interference by gas chromatography (Frey et al., 2020; Ji et al., 2015). We detected the nitrogen masses 44 (i.e., <sup>44</sup>N<sub>2</sub>O representing <sup>14</sup>N<sup>14</sup>N<sup>16</sup>O), 45 (i.e., <sup>45</sup>N<sub>2</sub>O representing <sup>14</sup>N<sup>15</sup>N<sup>16</sup>O or <sup>15</sup>N<sup>14</sup>N<sup>16</sup>O), and 46 (i.e., <sup>46</sup>N<sub>2</sub>O representing <sup>15</sup>N<sup>15</sup>N<sup>16</sup>O), and the isotope ratios 45/44, 46/44 with a GC-IRMS system (Delta V Plus, Thermo). Standards



in 20 mL glass vials with a known amount of  $N_2O$  gas were measured every two to three samples to calibrate for the  $N_2O$  concentration. The  $N_2O$  reference had the following isotopic composition:  $\delta^{15}N=-0.65\pm0.08$  % and  $\delta^{18}O=37.37\pm0.27$  % present in  $^{45}N_2O$  and  $^{46}N_2O$ . The total  $N_2O$  concentration and  $^{45}N_2O/^{44}N_2O$  and  $^{46}N_2O/^{44}N_2O$  ratios were converted to moles of  $^{44}N_2O$ ,  $^{45}N_2O$  and  $^{46}N_2O$ .  $N_2O$  production rates for each treatment were calculated from the slope of the increase in mass 45 and 46 during the linear phase over time. The  $N_2O$  production ( $R_{15-N_2O}$ , nmol-N L<sup>-1</sup> d<sup>-1</sup>) was calculated according to the following equation (1) (Santoro et al., 2020):

$$R_{15-N_2O} = (F_N)^{-1} \left( \frac{\Delta^{45}N_2O}{\Delta t} + 2 \frac{\Delta^{46}N_2O}{\Delta t} \times (F_N)^{-1} \right)$$
 (1)

where  $\Delta^{45}N_2O$  and  $\Delta^{46}N_2O$  represent the variation in the concentration of  $^{45}N_2O$  and  $^{46}N_2O$  over the incubation time ( $\Delta t$ ), and the  $F_N$  represents the fraction of  $^{15}N$  in the initial substrate pool ( $NH_4^+$  or  $NO_3^-$ ), which is assumed to be constant over the incubation time. The equation includes an extra factor of ( $F_N$ )-1 to account for the probability of  $^{46}N_2O$  production, which is proportional to ( $F_N$ )-2. Natural abundance 1000 ppm  $N_2O$  carrier gas (50  $\mu$ L in He) was injected before measurement to trap the produced labeled  $N_2O$  and to ensure a sufficient mass for isotope analysis.

## 2.6 <sup>15</sup>N-NO<sub>2</sub> production

After N<sub>2</sub>O analysis, we analyzed the samples incubated with <sup>15</sup>NH<sub>4</sub><sup>+</sup> and <sup>15</sup>NO<sub>3</sub><sup>-</sup> for <sup>15</sup>NO<sub>2</sub><sup>-</sup> production to determine the rates of NH<sub>4</sub><sup>+</sup> oxidation to NO<sub>2</sub><sup>-</sup>, and NO<sub>3</sub><sup>-</sup> reduction to NO<sub>2</sub><sup>-</sup> (first step of denitrification). The method is based on the isotopic analysis of N<sub>2</sub>O generated from the <sup>15</sup>NO<sub>2</sub><sup>-</sup>. Sample size was adjusted to contain 10 nmol of NO<sub>2</sub><sup>-</sup>, transferred into 20-mL glass vials, and purged with He for 10 min. The NO<sub>2</sub><sup>-</sup> was then converted to N<sub>2</sub>O using sodium azide in acetic acid (McIlvin and Altabet, 2005). During this reaction, one N from azide is transferred into the N<sub>2</sub>O molecule; hence the resulting values were corrected by multiplying by 0.5. The <sup>15</sup>N-N<sub>2</sub>O generated was measured on a Delta V Plus (Thermo) as described above. Net production rates of <sup>15</sup>NO<sub>2</sub><sup>-</sup> (R<sub>NO<sub>7</sub></sub>, nmol-N L<sup>-1</sup> d<sup>-1</sup>) were calculated following equation (2):

$$R_{NO_{2}^{-}} = \left(F_{NH_{4}^{+}}\right)^{-1} \frac{\Delta^{\left[15_{NO_{2}^{-}}\right]}}{\Delta t}$$
 (2)

where  $\Delta$ [ $^{15}NO_2$ -] represents the variation in the concentration of  $^{15}NO_2$ -,  $F_{NH_4^+}$  represents the fraction of  $^{15}NH_4$ - in the initial substrate pool, and  $\Delta t$  is the incubation time. Each rate was calculated from two time points, and two or three replicates per time point.

# 70 2.7 <sup>15</sup>N-NO<sub>3</sub> production


 $^{15}$ NO<sub>3</sub><sup>-</sup> production rate was measured by the increase in  $^{15}$ NO<sub>3</sub><sup>-</sup> in the samples incubated with  $^{15}$ NH<sub>4</sub><sup>+</sup> using the denitrifier method (Granger and Sigman, 2009; Sigman et al., 2001; Weigand et al., 2016). The method is based on the isotopic analysis of the N<sub>2</sub>O generated from the NO<sub>3</sub> by denitrifying bacteria that lack N<sub>2</sub>O-reductase activity (i.e., *Pseudomonas chlororaphis*). The  $^{15}$ N-N<sub>2</sub>O generated was measured as described above. We included known NO<sub>3</sub><sup>-</sup> isotope international standards (USGS34 and IAEA N3) and converted them to N<sub>2</sub>O using the denitrifier method to correct δ<sup>15</sup>N-N<sub>2</sub>O values. Net production of  $^{15}$ NO<sub>3</sub><sup>-</sup>



 $(R_{NO_3^-}, nmol-N L^{-1} d^{-1})$  is referred to here as nitrification (i.e., it includes the two-step process of oxidizing ammonium to nitrite to nitrate) and was calculated following equation (3):

$$R_{NO_3^-} = \left(F_{NH_4^+}\right)^{-1} \frac{\Delta \left[^{15}NO_3^-\right]}{\Delta t}$$
 (3)

where Δ[<sup>15</sup>NO<sub>3</sub>-] represents the variation in the concentration of <sup>15</sup>NO<sub>3</sub>-, F<sub>NH<sub>4</sub>+</sub> represents the fraction of <sup>15</sup>NH<sub>4</sub>+ in the initial substrate pool, and Δt is the incubation time. Each rate was calculated from two time points, and two or three replicates per time point.

## 2.8 Determination of N2O yields

The  $N_2O$  yield during  $NH_4^+$  oxidation to  $NO_2^-$  (Yield<sub>Amox</sub>, %) was defined as the percent of the total N transformed to  $N_2O$  during the incubation with  $^{15}N-NH_4^+$  (equation 4):

Yield<sub>Amox</sub> = 
$$\frac{R_{15-N_2O}}{R_{15-N_2O} + R_{NO_2^-}} \times 100$$
 (4)

The  $N_2O$  yield during nitrification (i.e.,  $NH_4^+$  oxidation to  $NO_3^-$ ) (Yield<sub>Nit</sub>, %) was defined as the percent of the total  $NH_4^+$  transformed to  $N_2O$  during the incubation with  $^{15}N-NH_4^+$  (equation 5):

$$Yield_{Nit} = \frac{R_{15-N_2O}}{R_{15-N_2O} + R_{NO_3^-}} \times 100$$
 (5)

The N<sub>2</sub>O yield during denitrification (Yield<sub>Denit</sub>, %) was calculated as follows (equation 6):

Yield<sub>Denit</sub> = 
$$\frac{R_{15-N_2O}}{R_{NO_2} + R_{15-N_2O}} \times 100$$
 (6)

## 2.9 Natural abundance of stable isotopes ( $\delta^{15}$ N and $\delta^{18}$ O)

Two serum bottles per depth were collected without headspace and killed with HgCl<sub>2</sub> to analyze the natural isotopic composition ( $\delta^{15}$ N) of the ambient pools of N<sub>2</sub>O, NO<sub>2</sub><sup>-</sup>, and NO<sub>3</sub><sup>-</sup>. A headspace was created with He before measuring the N<sub>2</sub>O, including standards with a known amount of N<sub>2</sub>O gas and internal standards for <sup>15</sup>N<sub>2</sub>O, as described before for the <sup>15</sup>N-N<sub>2</sub>O production rates. Both  $\delta^{15}$ N-N<sub>2</sub>O (‰) vs. Air-N<sub>2</sub> and  $\delta^{18}$ O-N<sub>2</sub>O (‰) vs. Vienna Standard Mean Ocean Water (VSMOW) were determined. Isotope measurements were linearity and offset corrected using an internal N<sub>2</sub>O reference gas with known isotopic composition (see above). Ideally, two known N<sub>2</sub>O reference gases would have been used for correction; however, due to this limitation, natural abundance isotope data were used to analyze trends in the sample dataset, rather than making comparison with previous studies. The natural isotopic composition of the NO<sub>2</sub><sup>-</sup>, and NO<sub>3</sub><sup>-</sup> pools (i.e.,  $\delta^{15}$ N-NO<sub>2</sub><sup>-</sup> and  $\delta^{15}$ N-NO<sub>3</sub><sup>-</sup>) were also determined in these samples by converting those compounds to N<sub>2</sub>O. NO<sub>2</sub><sup>-</sup> was converted to N<sub>2</sub>O by using the azide method (McIlvin and Altabet, 2005). We used the denitrifier method to convert NO<sub>3</sub><sup>-</sup> to N<sub>2</sub>O (Granger and Sigman, 2009; Sigman et al., 2001; Weigand et al., 2016). Both methods and corrections are described above.

## 2.10 Statistical tests

Statistical analyses were performed in R (R Core Team, 2014) version 4.4.0, including the Shapiro-Wilk test of normality analysis, the Levene's test for homogeneity of variance across groups, and the one-way analysis of variance test (ANOVA, F). The t-test (t) was used when the data were normally distributed, and the Welch t-test when the data were normally distributed but there was not homogeneity of variance across groups. When the data did not meet the assumptions of normality, we used the Kruskal-Wallis rank sum test (K-W) or the Wilcoxon test (W). We used the Grubbs test (G) to detect outliers.

## 3. Results






## 210 3.1 Dissolved $N_2O$ and other biogeochemical variables in the vertical profiles

The water column of Cubillas reservoir was thermally stratified in July (16.5 – 25.9 °C), such that oxygen varied dramatically with depth, with an oxygen peak at the top of the thermocline (>800 μmol L<sup>-1</sup>, 5.6 m) and decreasing concentrations until anoxia at 8 m (Fig. 1a). Dissolved N<sub>2</sub>O concentration increased from 0.11 in the epilimnion to 6.38 μmol-N L<sup>-1</sup> at the bottom of the reservoir. The decrease in the water level during the summer months due to human management presumably caused the mixing of the water column at the end of the summer, as evidenced in the homogenization of the thermal and oxygen profiles (Fig. 1a). Dissolved N<sub>2</sub>O distribution remained mostly homogeneous in September, ranging from 0.22 to 0.42 μmol-N L<sup>-1</sup> (Fig. 1a, Table S1). The water column was always supersaturated in N<sub>2</sub>O. NO<sub>3</sub><sup>-</sup> concentration decreased significantly from July to September (Fig. 1a, Table S1). The average NO<sub>3</sub><sup>-</sup> concentration was reduced by half, from 321.2 μmol-N L<sup>-1</sup> in July to 162.4 μmol-N L<sup>-1</sup> in September. NO<sub>2</sub><sup>-</sup> concentration varied from 13.8 to 33.0 μmol-N L<sup>-1</sup> (mean=22.0 μmol-N L<sup>-1</sup>). NH<sub>4</sub><sup>+</sup> concentration was below detection level at some depths, peaking at 4.3 and 6.9 μmol-N L<sup>-1</sup> in bottom waters. DOC concentrations varied from 217.6 to 247.7 μmol-C L<sup>-1</sup> (Table S1), and Chl *a* concentrations ranged from 5.4 to 18.1 μg L<sup>-1</sup> (Fig. 2).

Iznájar reservoir's water level decreased by over 5 m in summer, but thermal and oxygen stratification persisted due to its greater depth relative to Cubillas (Fig. 1b). The water column was always supersaturated in N<sub>2</sub>O (Table S1). Dissolved N<sub>2</sub>O increased with depth and over time, ranging from 0.05 to 0.26 μmol-N L<sup>-1</sup> in July, up to 3.60 μmol-N L<sup>-1</sup> in September, with the larger increase in the hypolimnion (Fig. 1b, Table S1). NO<sub>3</sub><sup>-</sup> concentration also decreased from July to September, from 373.7 to 329.3 μmol-N L<sup>-1</sup> (average values, Fig. 1b), with the lowest values at the oxycline, where NO<sub>2</sub><sup>-</sup> peaked. NH<sub>4</sub><sup>+</sup> was only detected in the oxycline in July and in the hypolimnion in September, with values of 5.7 and 8.7 μmol-N L<sup>-1</sup>, respectively. The DOC concentrations varied from 186.0 to 228.0 μmol-C L<sup>-1</sup>, and the Chl *a* concentrations from 3.8 to 12.4 μg L<sup>-1</sup> (Table S1, Fig. 3).

-10


0

 $\delta^{15}N-N_2O$  (%)

10

Figure 1. Physico-chemical profiles of Cubillas and Iznájar reservoirs. N<sub>2</sub>O concentration (μmol-N L<sup>-1</sup>, mean  $\pm$  standard error) and natural abundance (δ<sup>15</sup>N-N<sub>2</sub>O, ‰), water temperature (°C), DO concentration (μmol L<sup>-1</sup>), and the concentrations (μmol-N L<sup>-1</sup>) and natural abundances (δ<sup>15</sup>N, ‰) of NO<sub>3</sub><sup>-</sup>, NO<sub>2</sub><sup>-</sup> and NH<sub>4</sub><sup>+</sup> during July (orange) and September (purple) in Cubillas (a) and Iznájar (b) reservoirs. The dashed lines represent the suboxic zone (DO < 10 μmol L<sup>-1</sup>).

10

15 -50 25

 $\delta^{15}$ N-NO<sub>3</sub> (‰)  $\delta^{15}$ N-NO<sub>2</sub> (‰)





# 3.2 Distribution of N<sub>2</sub>O production and nitrification rates from <sup>15</sup>N-NH<sub>4</sub><sup>+</sup>

N<sub>2</sub>O production from NH<sub>4</sub><sup>+</sup> ranged from 0.06 to 48.57 nmol-N L<sup>-1</sup> d<sup>-1</sup> in the Cubillas reservoir (Fig. 2), and from 0.02 to 3.72 nmol-N L<sup>-1</sup> d<sup>-1</sup> in the Iznájar reservoir (Fig. 3) (n=12, Table S2). Ammonia oxidation rates (i.e., NO<sub>2</sub><sup>-</sup> production from NH<sub>4</sub><sup>+</sup>) were only significant in Iznájar's hypolimnion in September, reaching 215.8 ± 38.0 nmol-N L<sup>-1</sup> d<sup>-1</sup> (N<sub>2</sub>O yield=0.041%) (Table S2). In contrast, significant nitrification rates (i.e., NO<sub>3</sub><sup>-</sup> production from NH<sub>4</sub><sup>+</sup>) were detected at all study depths except in the hypolimnion of Iznájar in September (Figs. 2 and 3, Table S2). Nitrification rates varied from 6.1 to 56.1 μmol-N L<sup>-1</sup> d<sup>-1</sup> in Cubillas, and from 0.0 to 36.7 μmol-N L<sup>-1</sup> d<sup>-1</sup> in the Iznájar reservoir. The nitrification rates were significantly higher in July (mean±SD=24.6±19.4 μmol-N L<sup>-1</sup> d<sup>-1</sup>) than in September (7.3 ±6.7 μmol-N L<sup>-1</sup> d<sup>-1</sup>), and in Cubillas (mean±SD=22.2 ±17.9 μmol-N L<sup>-1</sup> d<sup>-1</sup>), than in the Iznájar reservoir (9.6 ±13.6 μmol-N L<sup>-1</sup> d<sup>-1</sup>) (p < 0.05, in both cases). The N<sub>2</sub>O yields during nitrification varied from 0.000 to 0.086 %, with the maximum yield observed in the bottom waters of Cubillas in July (Table S2). The production of N<sub>2</sub>O from NH<sub>4</sub><sup>+</sup> was significantly related to the *in situ* NH<sub>4</sub><sup>+</sup> concentration except in the hypolimnion of both reservoirs in September (n=10, adj R<sup>2</sup>=0.44, p < 0.05) (Fig. 4a). In these two samples we detected the highest NH<sub>4</sub><sup>+</sup> concentrations (>6 μmol L<sup>-1</sup>). The N<sub>2</sub>O production from NH<sub>4</sub><sup>+</sup> was an exponential function of the nitrification rates (Fig. 4b, adj R<sup>2</sup>=0.60, value<0.01).

## 3.3 Distribution of N<sub>2</sub>O production and NO<sub>3</sub> reduction rates from <sup>15</sup>N-NO<sub>3</sub>

 $N_2O$  production from  $NO_3^-$  varied from 0.2 to 18.1 nmol-N  $L^{-1}$  d<sup>-1</sup> in the Cubillas reservoir, and from 0.4 to 61.0 nmol-N  $L^{-1}$  d<sup>-1</sup> in the Iznájar reservoir (Figs. 2 and 3, Table S2). The highest rates were detected in the oxyclines.  $NO_3^-$  reduction to  $NO_2^-$  (i.e., first step of denitrification) varied from 13.7 to 33.2 µmol-N  $L^{-1}$  d<sup>-1</sup> in Cubillas, and from 10.1 to 28.6 µmol-N  $L^{-1}$  d<sup>-1</sup> in the Iznájar reservoir.  $NO_3^-$  reduction rates were significantly higher in July (27.5 ±7.0 µmol-N  $L^{-1}$  d<sup>-1</sup>) than in September (12.2 ±1.9 µmol-N  $L^{-1}$  d<sup>-1</sup>) (p < 0.05). This decrease in the  $NO_3^-$  reduction rates was coupled with a decrease in the  $NO_3^-$  concentration from July to September in both reservoirs. Among all the samples, the turnover time of  $NO_2^-$  (i.e.,  $NO_2^-$  concentration  $NO_2^-$  production by  $NO_3^-$  reduction varied from 0.2 days in the hypolimnion to 4.1 days in the oxycline of Iznájar in September (Table S2). The  $N_2O$  yield of  $NO_3^-$  reduction varied from 0.001 to 0.132 % in the Cubillas reservoir, and from 0.003 to 0.603 % in the Iznájar reservoir. The maximum yields occurred in the oxycline of Iznájar reservoir in September and the oxycline-bottom waters of Cubillas in September.  $N_2O$  production from  $NO_3^-$  was not significantly correlated to the *in situ*  $NO_3^-$  concentration (p=0.932).

## 3.4 In situ abundance of functional genes

The *in situ* abundance of the functional genes (archaeal *amoA*, *nirS* and *nosZ*) varied with depth, time, reservoirs, and with the N transformation rates (Figs. 2 and 3, Table S3). Archaeal *amoA* abundance ranged from 0 to 2.7 x 10<sup>3</sup> copies mL<sup>-1</sup> (n=12). In the Cubillas reservoir in July, the archaeal *amoA* gene was detected only in the oxycline, where NO<sub>2</sub><sup>-</sup> concentration was maximal and NH<sub>4</sub><sup>+</sup> minimal. We detected the archaeal *amoA* gene at all three depths in September, and its abundance decreased



with depth. In the Iznájar reservoir, the archaeal amoA gene was detected at all depths, with the minimum abundance in the oxycline in July. Archaeal amoA abundance wasn't related to the N<sub>2</sub>O concentration (p=0.85), the N<sub>2</sub>O production rates from NH<sub>4</sub><sup>+</sup> (p=0.139), or the nitrification rates (p=0.107).

The *nirS* abundance ranged from  $4.5 \times 10^4$  to  $5.3 \times 10^5$  copies mL<sup>-1</sup> in Cubillas, and from  $8.1 \times 10^4$  to  $4.7 \times 10^6$  copies mL<sup>-1</sup> in Iznájar (n=12). *nirS* was present in all the samples, and its abundance increased with depth and over time in Iznájar. The *nosZ* gene was only quantified in the deepest layers (n=4), where it ranged from 800 to  $2.1 \times 10^3$  copies mL<sup>-1</sup> and was higher in September than in July in both reservoirs. N<sub>2</sub>O production from NO<sub>3</sub><sup>-</sup> was not significantly related to the *in situ nirS* gene abundance (p=0.275).

Figure 2. Vertical profiles of the  $N_2O$  concentration, production rates, target genes (colored bars), and other relevant biogeochemical variables in the Cubillas reservoir in July (a) and September (b). Dissolved  $N_2O$  (µmol-N L<sup>-1</sup>, mean  $\pm$  standard error), and DO concentration (µmol L<sup>-1</sup>); Chl a concentration (µg L<sup>-1</sup>), and DIN concentration (µmol-N L<sup>-1</sup>);  $N_2O$  production (nmol-N L<sup>-1</sup> d<sup>-1</sup>) from  $NH_4^+$ ;  $N_2O$  production (nmol-N L<sup>-1</sup> d<sup>-1</sup>) and  $NO_2^-$  production (µmol-N L<sup>-1</sup> d<sup>-1</sup>) from  $NO_3^-$ , and the abundance of the target genes (x  $10^3$  copies mL<sup>-1</sup>, mean  $\pm$  standard deviation). The light gray area represents the suboxic zone (DO < 10 µmol L<sup>-1</sup>) and the dark grey the sediment. n.d. stands for not detected. Note the logarithmic scale. *nosZ* gene abundance was only determined in the deepest layers.

Figure 3. Vertical profiles of the N<sub>2</sub>O concentration, production rates, target genes (colored bars), and other relevant biogeochemical variables in the Iznájar reservoir in July (a) and September (b). Dissolved N<sub>2</sub>O (μmol-N L<sup>-1</sup>, mean ± standard error), and DO concentration (μmol L<sup>-1</sup>); Chl *a* concentration (μg L<sup>-1</sup>), and DIN concentration (μmol-N L<sup>-1</sup>); N<sub>2</sub>O production (nmol-N L<sup>-1</sup> d<sup>-1</sup>) and nitrification (NO<sub>3</sub><sup>-</sup> production, μmol-N L<sup>-1</sup> d<sup>-1</sup>) from NH<sub>4</sub><sup>+</sup>; N<sub>2</sub>O production (nmol-N L<sup>-1</sup> d<sup>-1</sup>) and NO<sub>2</sub><sup>-</sup> production (μmol-N L<sup>-1</sup> d<sup>-1</sup>) from NO<sub>3</sub><sup>-</sup>, and the abundance of the target genes (x 10<sup>3</sup> copies mL<sup>-1</sup>, mean ± standard deviation). The light gray area represents the suboxic zone (DO < 10 μmol L<sup>-1</sup>) and the dark grey the sediment. n.d. stands for not detected. Note the logarithmic scale. *nosZ* gene abundance was only determined in the deepest layers.



Figure 4. Drivers of N<sub>2</sub>O production from NH<sub>4</sub><sup>+</sup>. (a) Exponential relationship between the *in situ* NH<sub>4</sub><sup>+</sup> concentration (μmol-N L<sup>-1</sup>) and the N<sub>2</sub>O production rates (nmol-N L<sup>-1</sup> d<sup>-1</sup>), (b) relationship between the nitrification rates (nmol-N L<sup>-1</sup> d<sup>-1</sup>) and the N<sub>2</sub>O production. NH<sub>4</sub><sup>+</sup> concentrations > 6 μmol-N L<sup>-1</sup> are shown in lighter purple but excluded from the analysis in (a).

## 3.5 Changes in concentration and isotopic composition of N<sub>2</sub>O and inorganic nitrogen

Figure 1 and Table S4 illustrate depth distributions of DIN concentrations and isotopic compositions. Relationships between DIN concentrations and isotopic compositions are shown in Fig. 5. The natural abundance  $\delta^{15}$ N-N<sub>2</sub>O in the Cubillas reservoir ranged from -2.1 ‰ in the bottom waters in July to 3.6 ‰ in the epilimnion in September, while the  $\delta^{15}$ N-N<sub>2</sub>O in the Iznájar reservoir ranged from -8.7 ‰ in the hypolimnion in July to -2.3 ‰ in the hypolimnion in September (Figs. 1, 5). The  $\delta^{18}$ O-N<sub>2</sub>O ranged from 41.6 ‰ in the bottom waters of the Cubillas reservoir in July to 64.4 ‰ in the bottom waters of the Cubillas reservoir in September (Fig 5b,c).  $\delta^{15}$ N-NO<sub>3</sub><sup>-</sup> was consistently positive (i.e., <sup>15</sup>N enriched pool) in all the samples analyzed, and it varied from 8.9 to 13.4 ‰ (Fig. 5e). In the Iznájar reservoir, NO<sub>3</sub><sup>-</sup> concentration also decreased from July to September, along with an increase in  $\delta^{15}$ N-NO<sub>3</sub><sup>-</sup> (e.g., Fig 5e, #7-9). In the study reservoirs,  $\delta^{15}$ N-NO<sub>2</sub><sup>-</sup> varied more than  $\delta^{15}$ N-NO<sub>3</sub><sup>-</sup>. In general,  $\delta^{15}$ N-NO<sub>2</sub><sup>-</sup> increased with depth, showing changes in a few meters, from <sup>15</sup>N-depleted to <sup>15</sup>N-enriched values, except for the Iznájar reservoir in the July sampling (Fig. 1b).


Figure 5. Relationships between the concentrations of the dissolved  $N_2O$ ,  $NO_3^-$ , and  $NO_2^-$  (µmol-N L<sup>-1</sup>), and their natural isotopic 310 compositions. Note the logarithmic scales in the  $N_2O$  axis. The lines represent some trends mentioned in the Discussion. The ratio  $\delta^{18}O:\delta^{15}N = 2.5$  in (c) is indicative of active  $N_2O$  reduction (Ostrom et al., 2007).

## 3.6 Relationships between N<sub>2</sub>O concentration, production, and biogeochemical markers

In both reservoirs, the higher  $N_2O$  concentrations were found in the deepest layers under suboxic conditions (i.e., DO < 10 µmol  $L^{-1}$ ), and coincided with the highest cumulative Chl a concentration (mg Chl a m<sup>-2</sup>), and the highest abundances of *nirS* gene (Figs. 1-3).  $N_2O$  concentration decreased exponentially as DO concentration increased (Fig. 6a), but it increased in a power function related to cumulative Chl a concentration (Fig. 6b).  $N_2O$  concentration was also a power function of the *nirS* 

abundance (Fig. 6c). It is thus consistent that *nirS* abundance showed a negative relationship with DO concentration (Fig. 6d) and a positive correlation with cumulative Chl a concentration (Fig. 6e). Total production of N<sub>2</sub>O, calculated as the sum of the production from NH<sub>4</sub><sup>+</sup> and NO<sub>3</sub><sup>-</sup>, was significantly positively related to the *nirS* gene abundance (Fig. 6f, n=11).

Additionally, there was a positive correlation between  $\delta^{15}\text{N-NO}_3^-$  and the  $\delta^{15}\text{N-N}_2\text{O}$  (Fig. 6g). We also detected a strong relationship between  $\delta^{15}\text{N-NO}_2^-$  and N<sub>2</sub>O concentration (Fig. 6h). The abundance of the archaeal *amoA* gene was not related to  $\delta^{15}\text{N-NO}_2^-$  (p=0.99). In contrast,  $\delta^{15}\text{N-NO}_2^-$  was significantly related to the *nirS* abundance (Fig. 6i, n=12, adj R<sup>2</sup>=0.28, p < 0.05). Particularly, the *nirS* gene abundance explained up to 94% of the variance in  $\delta^{15}\text{N-NO}_2^-$  in the Iznájar reservoir (Fig. 6i, n=6, adj R<sup>2</sup>=0.94, p 


Figure 6. Drivers of dissolved N<sub>2</sub>O concentration and production. Dissolved N<sub>2</sub>O concentration (μmol-N L<sup>-1</sup>) as a function of (a) DO (μmol L<sup>-1</sup>); (b) cumulative Chl a concentration (mg Chl a m<sup>-2</sup>), and (c) nirS gene abundance (copies mL<sup>-1</sup>). nirS abundance as function of the (d) DO, and (e) cumulative Chl a concentration. (f) Total production of N<sub>2</sub>O (nmol-N L<sup>-1</sup> d<sup>-1</sup>) is a function of the nirS abundance. Note that sample #12 (Hypolimnion of Iznájar in September) in (f) is an outlier, and it was not included in the analysis. (g)  $\delta^{15}$ N-N<sub>2</sub>O as function of the  $\delta^{15}$ N-NO<sub>3</sub>- (‰), (h) dissolved N<sub>2</sub>O as function of the  $\delta^{15}$ N-NO<sub>2</sub>- (‰), and (i)  $\delta^{15}$ N-NO<sub>2</sub>- as function of nirS gene abundance. A second discontinuous trend line and equation have been drawn in (i) only for the Iznájar samples (n=6). Note the logarithmic scales in the x and y-axis. Correspondence between numbers and samples is shown in Table 1 and Figs. 2 and 3.

## 4 Discussion







N loading from the surrounding watershed significantly impacts the studied reservoirs, resulting in NO<sub>3</sub><sup>-</sup> concentrations exceeding 300 µmol-N L-1. The water columns of reservoirs have the capacity to process and remove significant amounts of N, as shown here through changes in DIN and N<sub>2</sub>O concentrations (Fig. 1), detection of N removal processes in <sup>15</sup>N isotope tracer experiments, presence of functional genes encoding the loss pathways (Figs. 2 and 3), and interpretation of patterns in natural abundance of N and O isotopes in the DIN and N<sub>2</sub>O pools (Fig. 5). NO<sub>3</sub><sup>-</sup> concentration decreased by 49% and 12% in Cubillas and Iznájar, respectively, in just two months, which represents a substantial net N loss. N removal processes also drive the production of the potent greenhouse gas N<sub>2</sub>O. The studied reservoirs had large accumulations of N<sub>2</sub>O in their deep waters, up to 6.38 μmol-N L<sup>-1</sup> in Cubillas reservoir in July, and up to 3.60 μmol-N L<sup>-1</sup> in Iznájar reservoir in September. During the study period, this accumulation of N<sub>2</sub>O in the water column of Cubillas and Iznájar reservoirs was affected by the water column depth and thermal stratification. Many reservoirs in the Mediterranean region are subject to significant evaporation during the summer and intense human management, resulting in substantial fluctuations in water level. Although both reservoirs experienced a decrease in water depth, this change affected the water column biogeochemistry only in the Cubillas reservoir, likely due to its smaller size. Use of the Cubillas reservoir caused a water-level drawdown from July to September that reduced the hydrostatic pressure and altered the water column stratification. Unstratified conditions exposed the high N<sub>2</sub>O deep waters to the reservoir surface, which likely led to a massive release of N2O both directly from the reservoir and, particularly, by degassing at the dam outflow or further downstream. The dam outflow is typically located at the oxyclinehypolimnion level, where the highest concentrations of greenhouse gases are found. Unfortunately, we were unable to quantify these N<sub>2</sub>O fluxes, but the concentration detected in bottom waters in July (6.38 μmol-N L<sup>-1</sup>, depth=9.5 m) versus September (0.42 µmol-N L<sup>-1</sup>, depth=6.2 m) suggests a massive release of N<sub>2</sub>O to the atmosphere during the summer. In contrast, the Iznájar reservoir did not lose thermal stratification from July to September and developed a steep oxygen gradient and an anoxic hypolimnion throughout the summer. N2O concentration increased throughout the water column during the summer, with the most significant increase occurring in the hypolimnion (1400% in the hypolimnion vs ~300% increase in the epilimnion and oxycline), which implies that N<sub>2</sub>O likely remains stored in that layer, and may be emitted during the fall mixing.

# 4.1 Active N<sub>2</sub>O production indicated by <sup>15</sup>N tracer incubations and functional genes

We detected significant production of N<sub>2</sub>O from both NH<sub>4</sub><sup>+</sup> and NO<sub>3</sub><sup>-</sup>. The rates of N<sub>2</sub>O production from NH<sub>4</sub><sup>+</sup> reported in this study are larger than those found in Lake Lugano (Frame et al., 2017) and closer to those detected in the Chesapeake Bay (Tang et al., 2022). These rates are also larger than the rates found in the eastern tropical South Pacific oxygen minimum zone (Frey et al., 2020; Ji et al., 2015). N<sub>2</sub>O production rates were significantly related to the availability of NH<sub>4</sub><sup>+</sup> and to nitrification rates, but not to the archaeal *amoA* gene abundance. Although the highest *amoA* abundance was measured in the oxycline of Cubillas in July (i.e., 2.7 x 10<sup>3</sup> copies mL<sup>-1</sup>), *amoA* was not detected in the surface and bottom waters within the same profile, precisely where the highest N<sub>2</sub>O production from NH<sub>4</sub><sup>+</sup> occurred. The absence of detectable archaeal *amoA* genes in samples




with high N<sub>2</sub>O production may reflect primer bias rather than true absence of ammonia-oxidizing archaea. Previous work in San Francisco Bay revealed that dominant AOA clades were not amplified by commonly used primers, including those employed in this study (Rasmussen and Francis, 2022). It is therefore possible that important AOA lineages present in these reservoirs were missed, leading to an underestimation of *amoA* abundance. We did not measure the bacterial *amoA* gene abundance, because AOA were the dominant ammonia-oxidizers in the study reservoirs (León-Palmero et al., 2023). Sample water was also pre-filtered before DNA extraction (pore size=3 μm). Therefore, microbes attached to particles or suspended sediment could not be assessed.

Significant nitrification rates were detected in 11 out of 12 samples, with values similar to those found in another eutrophic freshwater system, Lake Mendota (Hall, 1986), and several orders of magnitude higher than reported open ocean nitrification rates (e.g., 0.4 - 10 nmol-N L<sup>-1</sup> d<sup>-1</sup>) (Small et al. 2013, and references therein). The detection of high nitrification rates, but no significant ammonia oxidation, might suggest that comammox is occurring at these depths. However, our PCR analysis showed no evidence of the presence of comammox bacteria (Fig. S2). Instead, we hypothesize that the NO<sub>2</sub><sup>-</sup> production by ammonia oxidation was tightly coupled to NO<sub>2</sub><sup>-</sup> consumption by NO<sub>2</sub><sup>-</sup> oxidizers, such that it could not be detected in the NO<sub>2</sub><sup>-</sup> pool. NO<sub>2</sub><sup>-</sup> production from ammonia oxidation was only detected in one sample in which we did not detect a significant nitrification rate (i.e., hypolimnion of Iznájar reservoir in September, #12), suggesting that NO<sub>2</sub><sup>-</sup> could accumulate due to a decoupling of ammonia oxidation and nitrite oxidation in this sample. Ammonia oxidation is the rate-limiting step for nitrification in most systems, which is why NO<sub>2</sub><sup>-</sup> rarely accumulates in the environment and could explain our observed mismatch between ammonia oxidation rates and total nitrification rates (Kowalchuk and Stephen, 2001). The rates of NO<sub>3</sub><sup>-</sup> production detected here were often sufficient to account for a complete turnover of the NO<sub>2</sub><sup>-</sup> pool during the incubation, consistent with the idea that NO<sub>2</sub><sup>-</sup> did not accumulate, even though the in situ concentrations were substantial.

The production of N<sub>2</sub>O from NO<sub>3</sub><sup>-</sup> was generally higher than from ammonium, suggesting that NO<sub>3</sub><sup>-</sup> is the main substrate for N<sub>2</sub>O production. The highest rates occurred in oxycline samples, where NO<sub>3</sub><sup>-</sup> concentration was often lowest, and the NO<sub>2</sub><sup>-</sup> peaked. However, the N<sub>2</sub>O production from NO<sub>3</sub><sup>-</sup> was not significantly related to the *in situ* concentration of NO<sub>3</sub><sup>-</sup>, probably because N<sub>2</sub>O production rates are not limited by NO<sub>3</sub><sup>-</sup> availability. These rates were higher than the rates found in ocean waters (Ji et al., 2015), and in the Chesapeake Bay (Tang et al., 2022), but similar to those found in the eastern tropical South Pacific oxygen minimum zone (Frey et al., 2020). Similarly, these previous studies in oxygen minimum zones found the highest rates of N<sub>2</sub>O production close to the oxic-anoxic interface (Frey et al., 2020; Ji et al., 2015).

Denitrification is the main microbial process leading to NO<sub>3</sub><sup>-</sup> removal in aquatic systems. Denitrifying bacteria (as represented by the *nirS* gene) were consistently found throughout the reservoir water columns and reached their highest abundances in the suboxic waters. Their abundance was not significantly related to the N<sub>2</sub>O production from NO<sub>3</sub><sup>-</sup>, likely because of the small sample size (n=7). Frey et al. (2020) found that the *nirS* gene was not significantly correlated to N<sub>2</sub>O production from NO<sub>3</sub><sup>-</sup>, but was correlated with NO<sub>2</sub><sup>-</sup>. The total N<sub>2</sub>O production, calculated as the sum of the production from NH<sub>4</sub><sup>+</sup> and from NO<sub>3</sub><sup>-</sup> (Table S2), was significantly related to the *nirS* gene abundance (Fig. 6f), highlighting the importance of denitrification in the overall production of N<sub>2</sub>O. This is consistent with the higher production obtained from NO<sub>3</sub><sup>-</sup> than from NH<sub>4</sub><sup>+</sup>, and with the





evidence from natural abundance isotopes, discussed below. The rates of NO<sub>3</sub><sup>-</sup> reduction to NO<sub>2</sub><sup>-</sup> in this study were up to 1,000 times higher than those in the ocean (Füssel et al., 2012; Ji et al., 2015) and in the Chesapeake Bay (Tang et al., 2022). These eutrophic reservoirs exhibit high productivity, with elevated concentrations of NO<sub>3</sub><sup>-</sup> and organic matter fueling intense denitrification and N<sub>2</sub>O production. This rapid processing activity may reflect a system-level response to external nutrient loading, whereby a portion of the nitrogen input is redirected toward atmospheric release (León-Palmero, 2023).

## 4.2 Natural abundance stable isotopes support the role of denitrification

In general, microbial activity produces a significant isotopic fractionation of  $^{15}N$ , meaning that the lighter  $^{14}N$  is preferentially used in  $N_2O$  production, resulting in a  $N_2O$  pool relatively depleted in  $^{15}N$  relative to the respective substrate and a higher  $\delta^{15}N$  value in the substrate left behind (Wenk et al., 2013). In contrast, AOA produce  $N_2O$  that is enriched in  $^{15}N$  relative to the substrate, increasing  $\delta^{15}N$ - $N_2O$ , with an isotopic fractionation value of  $\approx$  -6 ‰ (Santoro et al., 2011; Stieglmeier et al., 2014). At the same time, the consumption of  $N_2O$  by denitrifiers increases the proportion of  $^{15}N$  and  $^{18}O$  in the remaining  $N_2O$  pool, increasing  $\delta^{15}N$ - $N_2O$  and  $\delta^{18}O$ - $N_2O$  values (Wenk et al., 2016).

To identify trends, and interpret them in relation to the processes that leave their signatures in the isotopes, each sample is identified on the cross plots with a unique number (Table 1 and Figs. 2, 3, 5, 6). The trends that we observed in the natural isotopic composition of the N species suggested that denitrification was a significant process in the water column, in agreement with the rate data. In general, the increase in the  $N_2O$  concentration with depth was coupled to the  $\delta^{15}N$ -N<sub>2</sub>O decrease (e.g., #1-3, #5-6 or #7-9 in Figs. 1 and 5a), which indicates net production of N<sub>2</sub>O. In contrast, the opposite trend occurred in Iznájar in September (#10-12, Figs. 1b and 5a), which suggests that N<sub>2</sub>O may be a mix of consumption by denitrifiers and production by AOA in the hypolimnion at the end of the summer. There was also an increase in the  $\delta^{18}O$ -N<sub>2</sub>O with depth in each profile, coupled with an increase in N<sub>2</sub>O concentration, which also suggests a parallel production and consumption of N<sub>2</sub>O at the deeper layers. That trend was not observed in Cubillas reservoir in July, but rather a noticeable increase in the  $\delta^{18}O$ -N<sub>2</sub>O in bottom waters from July to September along with N<sub>2</sub>O concentration decrease (Fig. 5b, #3 and #6), indicating active N<sub>2</sub>O reduction. Besides, many samples are located along the ratio  $\delta^{18}O$ : $\delta^{15}N = 2.5$  in Fig. 5c, which is indicative of active N<sub>2</sub>O reduction (Ostrom et al., 2007). We detected the *nosZ* gene, which encodes the reduction of N<sub>2</sub>O during denitrification, in hypolimnetic waters with higher abundances in September. N<sub>2</sub>O consumption can occur in the anoxic hypolimnion of Mediterranean reservoirs and result in undersaturations up to 27% in those with low N availability (León-Palmero et al., 2023).

However, in the investigated reservoirs, the N<sub>2</sub>O reduction by *nosZ*-carrying denitrifiers did not cause an undersaturation of N<sub>2</sub>O in the investigated time frame, which is consistent with previous findings in eutrophic reservoirs with high N availability (León-Palmero et al., 2023).

In the Iznájar reservoir, the decrease in  $NO_3^-$  concentration coincided with the increase in  $\delta^{15}N$ - $NO_3^-$ , suggesting that denitrification is consuming the lighter  $NO_3^-$  during these months (Fig 5e, #7-9). We detected that  $\delta^{15}N$ - $NO_3^-$  was correlated with  $\delta^{15}N$ - $N_2O$  (Fig. 6g), which is indicative of denitrification. Over time, as more  $N_2O$  is produced from  $NO_3^-$ , the  $NO_3^-$  pool may get substantially enriched in  $^{15}N$ , and  $\delta^{15}N$ - $N_2O$  values may also increase, creating a trend line where higher  $\delta^{15}N$ - $NO_3^-$ 







corresponds to higher  $\delta^{15}$ N-N<sub>2</sub>O values. In general, NO<sub>2</sub><sup>-</sup> reduction enriches <sup>15</sup>N in the remaining NO<sub>2</sub><sup>-</sup> pool, while the production of NO<sub>2</sub><sup>-</sup> may decrease its  $\delta^{15}$ N-NO<sub>2</sub><sup>-</sup>. In the study reservoirs, the production of N<sub>2</sub>O by denitrification may have enriched in <sup>15</sup>N the remaining NO<sub>2</sub><sup>-</sup> pool, as evidenced by the tight coupling between N<sub>2</sub>O concentration and  $\delta^{15}$ N-NO<sub>2</sub><sup>-</sup> (Fig. 6h) and the increase in the  $\delta^{15}$ N-NO<sub>2</sub><sup>-</sup> was coupled to the abundance of denitrifying bacteria in the reservoirs (Fig. 6i). The gene used as a marker for denitrifying bacteria (i.e., *nirS*) encodes the NO<sub>2</sub><sup>-</sup> reductase that catalyzes the reduction of NO<sub>2</sub><sup>-</sup> during denitrification. Thus, it acts directly on the NO<sub>2</sub><sup>-</sup> pool. Furthermore, the abundance of the *nirS* gene in the water column was correlated with the dissolved N<sub>2</sub>O, as we also detected in a survey of twelve Mediterranean reservoirs (León-Palmero et al., 2023). These results suggest that denitrification was the main pathway of N<sub>2</sub>O production, and it resulted in a characteristic isotopic imprint in the remaining NO<sub>2</sub><sup>-</sup> pool.

In addition, the cumulative Chl *a* concentration, which is a proxy for the vertical export of the autochthonous organic matter produced by primary producers in the whole water column, was significantly related to the abundance of the *nirS* gene and the dissolved N<sub>2</sub>O concentration (Fig. 6b,e). This is also consistent with our previous study in twelve reservoirs (León-Palmero et al., 2023), and may indicate that denitrification is enhanced by particulate material derived from the phytoplankton community. Several studies in marine waters have described that denitrification was affected by the quantity and quality of organic matter (Babbin et al., 2014; Ward et al., 2008). Dalsgaard et al. (2012) found that the higher denitrification rates were all found at marine stations with high Chl *a* levels in the overlying water, suggesting a subducted and potentially decaying algal bloom. In general, this organic matter export represents a high-quality carbon source, but also sinking particles with a surface for microbial colonization, an environment where both oxic and anoxic/low oxygen microenvironments coexist, and they even increase the probability of contact between bacteria and nitrogen (Liu et al., 2013; Xia et al., 2017).

#### 4.3 Implications for N<sub>2</sub>O concentration and fluxes

The highest total N<sub>2</sub>O production in Cubillas coincided with the highest N<sub>2</sub>O concentration at the deepest depth in both months. In the deeper reservoir, Iznájar, the highest production was measured at the oxycline, where there is a strong potential for N<sub>2</sub>O fluxes, while the highest N<sub>2</sub>O concentrations were detected in the hypolimnion. In both reservoirs, the N<sub>2</sub>O turnover time at the oxycline was the lowest in the profile. In Iznájar, the N<sub>2</sub>O turnover time at the oxycline was as low as 13 days in July and 8 days in September (Table S2), suggesting that the N<sub>2</sub>O produced at this location does not accumulate there. Instead, an important fraction of the N<sub>2</sub>O produced at the top of the oxycline may be consumed or diffuse to the top layer. This diffusive flux, together with the N<sub>2</sub>O produced *in situ* in the epilimnion by microbial activity and photochemidenitrification (Leon-Palmero et al., 2025), determines the large N<sub>2</sub>O fluxes found previously in this reservoir, reaching up to 3.6 mg N-N<sub>2</sub>O m<sup>-2</sup> d<sup>-1</sup>, and even exceeding the CO<sub>2</sub> equivalent warming potential from CO<sub>2</sub> and CH<sub>4</sub> emissions combined (León-Palmero et al., 2020a).





# 4.4 Scaling up to the reservoir level: how much nitrogen did the reservoirs lose?

An important feature observed in the water column of these reservoirs was the substantial decrease in the NO<sub>3</sub> concentration, suggesting an active N filter for the high N loadings. Microbial activity in the water column and the sediments of reservoirs can reduce the excess of N through emissions of N<sub>2</sub>, primarily produced during denitrification and anammox. In this study, N<sub>2</sub>O emissions also constitute an important loss of fixed N. Total DIN loss calculations from July to September showed that Cubillas lost 468 kg-N per day, while Iznájar lost 5337 kg-N per day, representing a 45 % and 11 % decrease, respectively (Table 2). The DIN loss rates (2.4 and 0.7 µmol-N L<sup>-1</sup> d<sup>-1</sup>) were similar or even higher than those calculated in other lakes or in the Baltic Sea (Seitzinger, 1988). Normalized to reservoir surface area, the N loss was slightly higher in Cubillas. N<sub>2</sub>O production was two orders of magnitude higher in Iznájar than in Cubillas in terms of kg-N per day, but production rates were more similar when normalized to area. In the water column of Iznájar, the percentage of the N<sub>2</sub>O production per DIN loss was higher than in Cubillas, at 1.9 % and 0.6 %, respectively. These percentages only refer to the biologically produced N<sub>2</sub>O in the water column and may increase if the N2O produced in the sediments, or the N2O produced abiotically by photochemodenitrification, which was initially described in the surface waters of these reservoirs (Leon-Palmero et al., 2025) are also incorporated in the calculation. Zhou et al. (2019) described a decrease of 97 % in the NO<sub>3</sub><sup>-</sup> concentration in the water column of Zhoucun reservoir during spring (2 months), and they related the N losses to aerobic denitrification occurring in the water column. Brezonik and Lee (1968) estimated that the hypolimnion of Lake Mendota lost 312 kg-N per day. Beaulieu et al. (2011) found that <1% of denitrified N was converted to N<sub>2</sub>O in streams. Thus, these reservoirs act as important sinks for fixed N during the summer at the landscape scale, particularly within agricultural and urban watersheds. Denitrification significantly contributed to N loss and N<sub>2</sub>O production in the water column. Although N<sub>2</sub>O production per unit of DIN loss was less than 2%, the absolute amount of N<sub>2</sub>O produced in the water column and likely emitted into the atmosphere is substantial.

**Table 2.** Total DIN loss, and  $N_2O$  produced from July to September in Cubillas and Iznájar reservoirs. Details on the calculations are provided in the Supplementary Material.

| Reservoir | Period |                       | DIN                                                                                      | loss                 | N <sub>2</sub> O production                                                         |    | N <sub>2</sub> O<br>production<br>per DIN loss |                                     |     |
|-----------|--------|-----------------------|------------------------------------------------------------------------------------------|----------------------|-------------------------------------------------------------------------------------|----|------------------------------------------------|-------------------------------------|-----|
|           | days   | Total,<br>μmol-N      | $\begin{array}{ll} \mu mol\text{-}N & L^{\text{-}1} \\ \text{d}^{\text{-}1} \end{array}$ | kg-N d <sup>-1</sup> | $\begin{array}{ll} \textbf{g-N} & \textbf{m}^{-2} \\ \textbf{d}^{-1} & \end{array}$ | %  | kg-N d <sup>-1</sup>                           | g-N m <sup>-2</sup> d <sup>-1</sup> | %   |
| Cubillas  | 64     | 2.1 x 10 <sup>6</sup> | 2.4                                                                                      | 468                  | 0.24                                                                                | 45 | 2.8                                            | 1.4 x 10 <sup>-3</sup>              | 0.6 |
| Iznájar   | 61     | $2.3 \times 10^7$     | 0.7                                                                                      | 5337                 | 0.20                                                                                | 11 | 101.5                                          | 3.9 x 10 <sup>-3</sup>              | 1.9 |

## 485 5 Conclusions




Our study shows that reservoir water columns actively process and remove N while producing  $N_2O$ , with denitrification as the dominant pathway. This is supported by changes in DIN and  $N_2O$  concentrations,  $^{15}N$  isotope tracer experiments, presence of functional genes, and patterns in natural abundance of N and O isotopes in the DIN and  $N_2O$  pools.  $N_2O$  was produced from both  $NH_4$  and  $NO_3$ , with higher rates from the latter, especially in oxycline layers. Total  $N_2O$  production, and concentration were significantly correlated with nirS gene abundance. In addition, nirS abundance and  $N_2O$  concentration were correlated with the cumulative Chl a concentration, suggesting that organic matter fuels intense denitrification and  $N_2O$  production. The patterns in natural abundance isotopes further support the predominance of denitrification.  $\delta^{15}N$ -NO $_3$  was positively correlated with  $\delta^{15}N$ -N $_2O$ , and  $\delta^{15}N$ -NO $_2$  increased with  $N_2O$  concentration and nirS abundance. Elevated  $\delta^{18}O$ -N $_2O$  and  $\delta^{18}O$ :  $\delta^{15}N$  ratio near 2.5, along with the detection of nosZ gene suggest active  $N_2O$  consumption in several layers, such as the hypolimnion of Iznájar reservoir. Cubillas showed the highest  $N_2O$  production and concentration at depth, likely followed by surface release during summer drawdown. In Iznájar,  $N_2O$  accumulated substantially in the hypolimnion over the summer, with peak production at the oxycline, where there is a strong potential for  $N_2O$  fluxes. Both reservoirs acted as substantial N sinks during the summer, losing 468 and 5337 kg-N per day, respectively. Therefore, the role of reservoirs as  $N_2O$  emitters should be characterized in more detail in future studies, especially considering their the global expansion and growing importance in  $N_2O$  budgets over the past century (Li et al., 2024; Wang et al., 2023).

#### Data availability

Data supporting the findings of this study are available within the article and in the Supplementary Material, which includes additional figures (Figs. S1 and S2), tables (Tables S1–S4), and detailed methodological descriptions (*DNA extraction, PCR and qPCR assays*, and *Scaling up to the reservoir level*).

#### 505 Author contribution

EL-P, CF and BBW designed the study, with inputs from RM-B, and IR. EL-P, RM-B, and IR contributed to data acquisition during the reservoir samplings. EL-P performed the experiments and processed the samples. All authors analyzed the data and discussed the results. EL-P wrote the first draft manuscript, which was complemented by significant contributions of all the authors.

## 510 Competing interests

The authors declare that they have no conflict of interest

## Acknowledgement

We are especially grateful to Eulogio Corral for his assistance in the field and to Alba Contreras-Ruiz for her support in the laboratory. We also thank Sergey Oleynik for his technical assistance with the mass spectrometer at Princeton University. We acknowledge the Hydrological Confederations of Guadalquivir (CHG) and the Andalusian Environmental and Water Agency (AMAYA) for facilitating the reservoir sampling and databases.

## Financial support

This research was funded by the Spanish Ministry of Science, Innovation and Universities, grant RTI2018- 098849-B-I00 (IR, RM-B) and co-financed with the grant PID2022.1378650B.100 funded by MICIU/AEI/10.13039/501100011033/ and by ERDF, EU. E.L-P was supported by a PhD fellowship from the Spanish Ministry of Education, Culture and Sport (grant nos. FPU014/02917), and later by a Marie Skłodowska-Curie postdoctoral fellowship (HORIZON-MSCA-2021-PF-01, project number: 101066750) by the European Commission at Princeton University. Additional financial support for EL-P's visits to Princeton University during her PhD in 2017 and in 2018 was provided by the Award for Excellence in International Student Mobility by the University of Granada, and the grant for Short-Term Mobility Support (EST17/00087) from the Ministry of Education, Culture and Sports, respectively.

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
