# Peer review of "Denitrification as the predominant process in nitrous oxide production in the water column of two eutrophic reservoirs"

_EGUsphere, 2025_

## Author Comment (AC1)

**Please see below the point-by-point responses to Referee 1, and the actions taken regarding their concerns.**

**Major changes**

The revised manuscript includes relevant changes with respect to the submitted manuscript in order to address the reviewer's concerns. Please note that, due to the reorganization of certain sections, subsection numbering and figure references have been updated accordingly in the revised manuscript. In addition, we have replaced "predominant" with "dominant" in the title for grammatical correctness. In the text that follows, the suggestions and comments of the Referees are in black and plain font, and our responses are in *italics and blue font.*

**Referee 1**

**General comment:**

In this manuscript, the authors report nitrous oxide production pathways in two eutrophic reservoirs using [15]N tracer incubations, natural-abundance isotopes, and functional gene quantification. The methodologies are well-established and proven effective in clarifying nitrous oxide production in aquatic systems; the datasets add valuable observations to the community; and the analyses generally support the claim that denitrification dominates water-column nitrous oxide production. The manuscript is suitable for target journal when properly revised. Note: line numbers and pages from authors' PDF file.

My main concern is in the title. The authors argue that denitrification could be the predominant process regulating nitrous oxide production, and in the Discussion section, the authors present evidence about higher nirS abundance and isotopic patterns associated with denitrification. However, this argument may be weakened by the fact that, denitrification was generally absent at the oxygenated surface water (figs. 2 & 3), and the $N_2O$ consumption potential inferred from natural abundance isotopic data. The authors acknowledged that *nosZ* was quantified only at the deepest depths (n=4) and therefore cannot constrain $N_2O$ reduction within the entire reservoir (line 270–275; p.12 figure caption). I suggest the authors reframe the argument/statement to better characterize the novelty of their work.

*We thank the reviewer for their thoughtful comment. We believe there may be a misunderstanding regarding the treatments applied during the incubation experiments. Specifically, the incubations with $^{15}NO_3^-$ to determine $N_2O$ production by denitrification were not conducted in the epilimnion; rather, they were performed in the oxycline and hypolimnion. This was described in the Methods section (lines 173-174) and summarized in Table 1.*

*"The first treatment ($^{15}N-NH_4^+ + {}^{14}N-NO_3^-$) was performed at all the depths (n=12), but the second treatment ($^{15}N-NO_3^- + {}^{14}N-NH_4^+$) was performed only at the oxycline and hypolimnion (n=7, Table 1)."*

*To avoid confusion, we have updated the figures to clearly indicate that the treatment with $^{15}NO_3^-$ used to detect denitrification was not performed in the epilimnion (see Fig. 2 below as an example, which corresponds to Fig. 3 in the revised manuscript). Therefore, we cannot conclude that denitrification was absent in oxygenated surface waters, only that it was not measured there.*

*As the reviewer correctly noted, natural abundance data suggest that $N_2O$ may also be consumed by denitrification, and we detected nosZ in the analyzed samples. However, despite*

*potential N₂O consumption, the overall effect of denitrification remains net production, as evidenced by the substantial accumulation of N₂O in the water column. We are aware that based on those results, we cannot constrain the N₂O consumption in the entire reservoir but can identify the net production trends.*

*In order to reframe the argument/statement and highlight the novelty of this work, we have done the following changes:*

*Abstract:*

*"Reservoirs are important sites for nitrogen cycling and a significant global source of the potent greenhouse gas nitrous oxide ($N_2O$). They receive nitrogen inputs from agriculture and urban sources, fueling $N_2O$ production via nitrification, denitrification, and photochemodenitrification. However, existing estimates of $N_2O$ production in reservoirs remain uncertain because most studies have focused on $N_2O$ in rivers or lake sediments, often overlooking the water column of lentic systems. Here, we present the first integrated assessment of $N_2O$ production pathways in reservoir water columns using stable isotope tracer incubations alongside analyses of in situ natural abundance of nitrogen pools and functional genes involved in nitrification (amoA) and denitrification (nirS), across two eutrophic reservoirs with contrasting morphometries. We used $^{15}N\text{-}NH_4\text{+}$ and $^{15}N\text{-}NO_3^-$ tracers to quantify rates of $N_2O$ production, nitrification, and nitrate reduction at the beginning and the end of the stratification period. Notably, nitrate concentration decreased by up to 49 % over the two months. $N_2O$ production from ammonium ranged from 0.02 to 48.6 $nmol\text{-}N\ L^{-1}\ d^{-1}$, while $N_2O$ from nitrate varied from 0.2 to 61.0 $nmol\text{-}N\ L^{-1}\ d^{-1}$. High rates of nitrification, nitrate reduction to nitrite, and rapid nitrite turnover were observed, with total $N_2O$ production significantly correlated with nirS gene abundance. A strong positive correlation was found between $\delta^{15}N\text{-}NO_2^-$ and both $N_2O$ concentration and nirS abundance. These findings reveal that denitrification and nitrite dynamics play a central role in $N_2O$ formation within reservoir water columns, advancing understanding of nitrogen loss and greenhouse gas emissions from lentic systems."*

*Introduction (Lines 66 – 76 in the revised manuscript):*

*"Shallow systems tend to emit $N_2O$ continuously due to weak thermal stratification and less capacity to accumulate $N_2O$. Further studies on $N_2O$ production in the water column of reservoirs with different morphometries are required to improve our knowledge of $N_2O$ emissions. To address this gap, we present the first integrated assessment of $N_2O$ production pathways in reservoir water columns, combining stable isotope tracer incubations with analyses of in situ natural abundances of the N pools and functional genes involved in $N_2O$ cycling to quantify $N_2O$ production rates and trace the origin of the $N_2O$ in the water column of two reservoirs. We used $^{15}N\text{-}NH_4^+$ to quantify the rates of $N_2O$ production from $NH_4^+$, and ammonia oxidation to nitrite and nitrate; and $^{15}N\text{-}NO_3^-$ to trace the formation of $N_2O$ and $NO_2^-$ from $NO_3^-$ reduction. Incubations were performed at three depths at the beginning and end of summer stratification. We selected a shallow and a deep reservoir (Cubillas and Iznájar, respectively) located in watersheds with high N inputs, both of them monomictic with significant emissions and concentrations of $N_2O$ (León-Palmero et al. 2020a, 2023), providing an ideal setting to explore $N_2O$ cycling."*

*Figure 2 revised (Figure 3 in the revised manuscript):*

[Figure]

**Figure 3.** Vertical profiles of the N₂O concentration, production rates, marker genes (colored bars), and other relevant biogeochemical variables in the Cubillas reservoir in July (a) and September (b). Dissolved N₂O (µmol-N L⁻¹, mean ± standard error), and DO concentration (µmol L⁻¹); Chl *a* concentration (µg L⁻¹), and DIN concentration (µmol-N L⁻¹); N₂O production (nmol-N L⁻¹ d⁻¹) and nitrification (NO₃⁻ production, µmol-N L⁻¹ d⁻¹) from NH₄⁺; N₂O production (nmol-N L⁻¹ d⁻¹) and NO₂⁻ production (µmol-N L⁻¹ d⁻¹) from NO₃⁻, and the abundance of the target genes (x 10³ copies mL⁻¹, mean ± standard deviation). Numbers next to N₂O concentrations refer to the sample ID in Table 1. The light gray area represents the suboxic zone (DO < 10 µmol L⁻¹) and the dark grey the sediment. <LOD means below level of detection. Note the logarithmic scales for some panels. *nosZ* gene abundance was only determined in the deepest layers. N₂O and NO₂⁻ production were only determined in the oxycline and hypolimnion.

**Some minor comments below:**

Line 31 – 33: These global estimates of increased N2O emissions from inland waters are often with large uncertainties. Better to state as "mean ± uncertainty"

*We agree with the reviewer, but unfortunately, the original article does not provide such uncertainties. Please see the original article by Wang et al. (2023)*

*Original text from Wang et al. (2023): "The increase in reservoir emission (0.44 Tg N yr$^{-1}$) during 1900–2010 accounts for 50% of the total increase in inland-water emission (0.89 Tg N yr$^{-1}$), indicating that reservoirs are currently an important source of N$_2$O to the atmosphere".*

*Wang, J., Vilmin, L., Mogollón, J. M., Beusen, A. H. W., van Hoek, W. J., Liu, X., Pika, P. A., Middelburg, J. J., and Bouwman, A. F.: Inland Waters Increasingly Produce and Emit Nitrous Oxide, Environ. Sci. Technol., 57, 13506–13519, https://doi.org/10.1021/acs.est.3c04230, 2023.*

Line 35 – 37: There are increasing number of literatures about GHG emission from reservoirs (doi: 10.5194/bg-11-5245-2014; 10.1016/j.watres.2025.123420; 10.3390/su132111621)

*We thank the reviewer for their suggestion. We have included the finding by Chen et al., (2025) in the introduction. In Sturm et al., (2014), no production was detected during sediment incubations. In Ion and Ene (2021) authors focus the GHG assessment on CO$_2$ and CH$_4$ emissions, so this paper does not provide useful background information about our subject, N$_2$O.*

*Please see the revised text below: "A recent study estimated that N2O accounted for more than 80% of the total GHG emissions from hydroelectric reservoirs in China in 2020 (Chen et al., 2025)." Lines 41 and 42 in the revised manuscript.*

Line 52 – 54: Should clarify about low but not zero oxygen promoting partial denitrification and thus net N2O production.

*We included that information in the introduction (Lines 55 – 60 in the revised manuscript):*

*"Denitrification is an anaerobic pathway, and oxygen regulates the activity of the denitrifying enzymes, especially the N$_2$O reductase (Bonin et al., 1989; Zumft, 1997). Therefore, at low but non-zero oxygen concentrations, N$_2$O reductase might be inhibited, promoting partial denitrification and resulting in net N$_2$O production. Moreover, many bacteria can denitrify in both oxic and anoxic conditions (Hochstein et al., 1984; Lloyd et al., 1987), and the presence of denitrifying bacteria has been demonstrated in the oxic and anoxic water column of lakes (Junier et al., 2008; Kim et al., 2011; Pajares et al., 2017) and reservoirs (León-Palmero et al., 2023)."*

Line 103: Filtrate passing through 0.7 μm GF/F filter may not be suitable to characterize DOC; 0.45μm filter is recommended.

*Water was filtered through pre-combusted (450 °C for 3 hours) Whatman GF/F glass-fiber filters with a nominal pore size of 0.7 μm, which after combustion have pore size equivalent to 0.45 μm.*

Line 110 – 120: Should mention the pore size of filter collecting molecular samples. Because genetic materials were collected from water pre-filtered through 3 μm, there may be bias toward underestimating particle-attached nitrifiers or denitrifiers (as stated line 370 – 371).

*After the pre-filtration step (3 µm), samples were not collected on a second filter but instead concentrated by centrifugation, following the procedure developed by Boström et al. (2004). This approach is described in detail in the Supplementary Material (Extended Methods: DNA extraction, PCR and qPCR assays).*

*We have explained that in the main material (Lines 149 - 151):*

*"We pre-filtered water samples through 3 µm pore size filters, and concentrated the samples by centrifugation, then extracted DNA following Boström et al. (2004), and applied PCR and qPCR to assess presence, and abundance of target genes."*

*We agree with the reviewer that pre-filtration through a 3 µm pore-size filter may bias toward underestimating particle-attached microorganisms. We now explicitly acknowledge this limitation in the Discussion, stating:*

*"Additionally, sample water was pre-filtered before DNA extraction (pore size = 3 µm), which may have excluded microbes attached to particles or suspended sediment, potentially including AOA or Comammox groups." Lines 413 – 415 in the revised manuscript.*

Line 122: It is better to specify the criteria for selecting the three depths in the main text. And specify when the N2O concentration or isotope samples were measured at which facility. Preservation using mercuric chloride is generally not recommended for sample storage longer than 1 year.

*We selected three depths representing the epilimnion, oxycline and hypolimnion or bottom waters of each system. This is explained in the Method **section 2.2 Vertical profiles and Biogeochemical characterization**:*

*"First, we conducted a vertical profile of the water column using a Sea-Bird 19plus CTD profiler, obtaining continuous measurements of temperature (ºC), dissolved oxygen (DO, µmol $L^{-1}$), and conductivity (µS $cm^{-1}$) in the reservoirs. Based on the temperature and DO profiles, we sampled three depths representing the epilimnion, oxycline, and hypolimnion or bottom waters. Water was collected at these three depths using a 5-L UWITEC bottle for further analyses and incubation experiments" Lines 96 – 101 in the revised manuscript.*

*In the revised manuscript, we have included the information required by the reviewer as follows:*

*a) **$N_2O$ concentration, Method section 2.2 Vertical profiles and Biogeochemical characterization (Lines 101-104 in the revised manuscript)***

*"Samples for dissolved $N_2O$ analysis were taken in 250-mL air-tight Winkler bottles in duplicate, preserved with a solution of $HgCl_2$ (final concentration 1 mmol $L^{-1}$) to inhibit biological activity, and sealed with Apiezon® grease to prevent gas exchange. Samples were stored in the dark at a controlled temperature (25 ºC) for less than six months until analysis at the University of Cádiz."*

*b) **Tracer incubations, Method section 2.5 Experimental setup of $^{15}N$ tracer incubations (Lines 176 and 177 in the revised manuscript)***

*"All samples were stored at room temperature in the dark for less than six months and shipped to the laboratory at Princeton University for further analysis."*

*c) **Isotopes, Method section 2.3 Natural abundance of stable isotopes ($\delta^{15}N$ and $\delta^{18}O$) (Lines 121 and 122 in the revised manuscript)***

*"Two serum bottles per depth were collected without headspace and poisoned with $HgCl_2$ to analyze the natural isotopic composition ($\delta^{15}N$) of the ambient pools of $N_2O$, $NO_2^-$, and $NO_3^-$.*

*Samples were maintained in darkness at room temperature for under six months before shipment to Princeton University for analysis.*"

Line 124: What was the volume of air headspace during oxic incubation?

*The headspace during the oxic incubation was similar to the anoxic one (=3 mL). This detail has been included in the revised manuscript, as follows:*

*"Once in the lab, samples from oxic water depths (refer to Table 1) were purged uncapped for 2 min to remove excess $N_2O$, and a  3-mL  headspace with ambient air was maintained after being exposed to ambient air for 30 min." Lines 161 – 163 in the revised manuscript.*

Line 152: The equation (1) by Santoro et al. will overestimate the N2O production from $^{46}N_2O$ signal. The equation proposed by Ji et al., 2018 GBC (doi: 10.1029/2018GB005887) is recommended.

*We thank the reviewer for this suggestion. Both equations (Santoro et al. and Ji et al., 2018) have been published, previously used and are valid for estimating $N_2O$ production. We chose the equation by Santoro et al. because it incorporates the additional factor of 1/F, which is essential for the 46 $N_2O$ term. This factor accounts for the probability that two $^{15}N$ atoms will pair to form $46N_2O$, which is proportional to $1/F^2$. Including this correction ensures a more accurate representation of the isotopic pairing process.*

*Similarly, other studies have applied the $1/F^2$ correction to account for the probability of forming doubly labeled molecules, such as $^{30}N_2$ in isotope pairing techniques (e.g., Thamdrup and Dalsgaard, 2002). This adjustment ensures accurate calculation of production rates because the likelihood of two $^{15}N$ atoms pairing is proportional to $F^2$.*

Line 186: It seems the incubation timepoints for nitrification were different from those of N2O production; which two time points were analyzed for 15N-nitrate?

*We used the four time points to study $N_2O$ production from 15N-Ammonia and from 15N-Nitrate ($t_0$, $t_1$, $t_2$, $t_3$), but we used only the first two time points to measure the rates of nitrification and nitrate reduction to nitrite. We have clarified this in the method section:*

a) **Method section, 2.6 $^{15}N$-$N_2O$ production rates from $^{15}NH_4^+$ and $^{15}NO_3^-$ (Lines 184 and 185 in the revised manuscript)**

*"$N_2O$ production rates for each treatment were calculated from the slope of the increase in mass 45 and 46 during the linear phase over the four time points."*

b) **Method section, 2.7 $^{15}N$-$NO_2^-$ production & 2.8 $^{15}N$-$NO_3^-$ production (Lines 202-203, and 217-218 in the revised manuscript).**

*"Each rate was calculated from the first two time points, and two or three replicates per time point."*

Line 183 – 190: This section 2.8 about calculating N2O yields using two equations, (4) & (5); which one is used to represent data described in line 240 & 246?

*We used Equation 4 to calculate the yield reported in Line 240 because it refers to the yield during ammonia oxidation. In contrast, the yields presented in Line 245 were calculated using Equation 5, as they correspond to the yield during nitrification. To make this information clearer*

*to the reader, we have renamed the yields in the method section and in the main text as $N_2O$-yield$_{Amox}$, $N_2O$-yield$_{Nit}$, and $N_2O$-yield$_{Denit}$.*

*Line 240 (285 in the revised manuscript): "Ammonia oxidation rates (i.e., $NO_2^-$ production from $NH_4^+$) were only significant in Iznájar's hypolimnion in September, reaching 215.8 ± 38.0 nmol-N $L^{-1}$ $d^{-1}$ ($N_2O$-yield$_{Amox}$=0.041 %) (Table S2)."*

*Line 245 (Line 292 in the revised manuscript): "The $N_2O$ yields during nitrification ($N_2O$-yield$_{Nit}$) varied from 0.000 to 0.086 %, with the maximum yield observed in the bottom waters of Cubillas in July (Table S2)."*

Line 191 – 202: Authors should clarify the consistent amount of N injected into mass spectrometry to determine natural abundance. This is important because the concentrations of N species varied with depth. If varying amounts of N were injected, even for the same water sample, mass spectrometry will yield varying isotopic values.

*We appreciate the reviewer's comment. To clarify, we used the same sample volume for all natural abundance measurements (a 60 mL serum vial after 3-mL of headspace was created).*

a) *To measure the natural isotopic composition of the $N_2O$, we did not adjust the amount of N injected into mass spectrometry but corrected for the mass effect by including standards with a known amount of $N_2O$ gas and internal standards for $^{15}N$-$N_2O$, as described in the method section. Both $\delta^{15}N$-$N_2O$ (‰) vs. Air-$N_2$ and $\delta^{18}O$-$N_2O$ (‰) vs. Vienna Standard Mean Ocean Water (VSMOW) were determined. Isotope measurements were linearity and offset corrected using an internal $N_2O$ reference gas with known isotopic composition (this information was detailed in the methods). Please see subsection 2.3 Natural abundance of stable isotopes ($\delta^{15}N$ and $\delta^{18}O$) in line 119 in the revised manuscript.*

b) *To measure the natural isotopic composition of the $NO_2^-$, and $NO_3^-$ pools (i.e., $\delta^{15}N$-$NO_2^-$ and $\delta^{15}N$-$NO_3^-$) we converted those compounds to $N_2O$. $NO_2^-$ was converted to $N_2O$ by using the azide method (McIlvin and Altabet, 2005). We used the denitrifier method to convert $NO_3^-$ to $N_2O$ (Sigman et al., 2001; Granger and Sigman, 2009; Weigand et al., 2016). Both methods and corrections are described in the Method section (2.3 Natural abundance of stable isotopes ($\delta^{15}N$ and $\delta^{18}O$, line 119 in the revised manuscript). During the azide method, the sample size was adjusted to contain 10 nmol of $NO_2^-$, and 20 nmol nitrate $NO_3^-$. We included known isotope international standards (USGS34 and IAEA N3).*

Line 212: Apparently high DO at the surface (> 400 micromolar, ~2-fold saturation) suggests a very strong oxygen source that can only be supported by blooming algal activity. This was not evident from the chl-a concentration profiles (10 – 20 microgram per liter)

*Thank you for catching that plotting error, which we have corrected in the revised manuscript. The value in the surface was about 250 µmol $O_2$ $L^{-1}$, which is about 100% saturation, but peaked at 400 µmol $O_2$ $L^{-1}$ at 5.6 m (Cubillas in July). Line 242 in the revised manuscript.*

Line 318–323: The main text should clarify "sample #12 excluded as specified in the caption of Fig. 6f" (as stated in caption L325–333).

*We have specified that sample #12 was excluded of the analysis in the revised main text. Plesae see line 352 in the revised manuscript.*

Line 335 & 466: I am not sure about the statement "remove significant amount of N". There could be possibility about the loss of nitrate due to algal assimilation, and the biomass being exported to the

sediment. In addition, statements about **reservoir-level DIN loss** described in Section 4.4 and the supplementary lack uncertainty estimates and assumptions (depth weighting, temporal representativeness, varying hydraulic retention time, etc). Please add a short paragraph enumerating assumptions and explain the possible caveats. Readers may realize that these DIN loss potential could be specific to the sampling period (Jul–Sep), not necessarily annual rates (L465–474 & Table 2).

*Thank you for this observation. We agree that nitrate loss in the water column can occur through multiple pathways, including algal assimilation and subsequent sedimentation of organic matter. Our statement "remove significant amount of N" was intended to refer broadly to fixed nitrogen removal from the water column, not exclusively to denitrification. To clarify, we have revised the text to acknowledge these alternative mechanisms. The revised sentence now reads:*

*"$NO_3^-$ concentration decreased by 49 % and 12 % in Cubillas and Iznájar, respectively, in just two months, which represents a substantial net N loss. This net loss in the water column likely reflects a combination of processes, including denitrification, algal assimilation followed by sedimentation of organic matter, and other biogeochemical transformations." Lines 371 -374.*

*We agree with the reviewer that the assumptions made during these calculations should be better stated. Therefore, we included the following paragraph in the section **Extended Methods: Scaling up to the reservoir level***

*"The following calculations rely on several assumptions. First, we assume that the July and September profiles are representative of the entire reservoir water column during the stratification period and that depth-weighted concentrations capture vertical variability. Second, we use average reservoir volume to approximate water outflow, without accounting for short-term fluctuations in hydraulic retention time or drawdown dynamics. Third, we assume minimal nitrogen inputs from the watershed during the study period, since summer is the dry period and observed reservoir drawdown (and thus minimal inputs from streams, rain or runoff) but cannot exclude minor contributions from groundwater or episodic events. Finally, these estimates reflect net changes over the sampling interval (July–September) and should not be interpreted as annual rates. Spatial heterogeneity and biological processes such as algal assimilation and sedimentation may also influence apparent DIN loss. Therefore, these values represent potential nitrogen removal only during the stratified season, not whole-year budgets."*

*Additionally, we also included the following paragraph in the revised discussion (Lines 525 – 530):*

*"These estimates represent a major seasonal N loss event rather than annual rates. They are based on DIN concentration differences between July and September, without considering whether the reservoirs received N inputs from their watersheds during that period. Since summer is the dry period, and drawdown of the reservoirs exceeded any input via rain or runoff, N inputs from the watersheds were likely minimal during the study period. Further details on the calculations and assumptions are provided in the Supplementary Material (Extended Methods: Scaling up to the reservoir level)."*

Line 362 – 365: Authors should clarify the inconsistencies about anoxic depths with measurable amoA gene abundance (fig 3b), or ammonium oxidation rates (fig 2a).

*We have discussed these apparent inconsistencies in the revised discussion (Lines 393 to 399 in the revised manuscript). Please see lines 388 – 404 in the revised manuscript:*

*"N₂O production rates were significantly correlated with the availability of NH₄⁺ and with nitrification rates, but not with archaeal amoA gene abundance. Despite the hypolimnion of Iznájar in September (#12) being apparently anoxic, we detected a significant production of N₂O from NH₄⁺, ammonia oxidation, and the presence of archaeal amoA genes. This combination of processes and gene detection suggests that trace amounts of oxygen may have been present at levels below the detection limit of our oxygen sensor. Similarly, the presence of trace levels of oxygen may explain the production of N₂O from NH₄⁺, and the nitrification rates in the anoxic waters of Cubillas, although in that case we did not detect the presence of archaeal amoA genes"*

Line 374 – 375: Where are the data for ammonia oxidation/nitrification, that can infer comammox?

*Initially, we hypothesized that comammox may be relevant in these reservoirs because we detected significant rates of nitrification (complete oxidation of ammonia to nitrate) but we rarely detect any significant rate of ammonia oxidation (ammonia to nitrite). These results are described in the subsection **3.2 Distribution of N₂O production and nitrification rates from ¹⁵N-NH₄⁺** . Then, comammox amoA genes were targeted in PCR assays using degenerate PCR primers for clades A and B (Pjevac et al., 2017), but we did not detect them. During these PCR assays, no positive control could be used in these assays, so we cannot completely rule out the presence of these bacteria. This is discussed in the revised text (418 – 423):*

*"The detection of high nitrification rates, but no significant ammonia oxidation, might suggest that comammox is occurring at these depths. However, our PCR analysis showed no evidence of the presence of comammox bacteria (Fig. S2), although because no positive control was available, we cannot completely exclude their presence. Therefore, we consider the possibility that complete ammonia oxidation could contribute to the observed nitrification rates. Alternatively, we hypothesize that the NO₂⁻ production by ammonia oxidation was tightly coupled to NO₂⁻ consumption by NO₂⁻ oxidizers, such that it could not be detected in the NO₂⁻ pool."*

Line 444: Are the authors suggesting denitrification potential in anoxic water column or the sediment? Please clarify.

*In that context, we suggest that denitrification in the water column may be enhanced by the export of organic matter, however, we acknowledge that such export can also stimulate denitrification in the sediment. Although our study focuses on processes occurring in the water column, we recognize that denitrification and/or fixed nitrogen loss in the sediments cannot be ruled out.*

*In the revised manuscript (lines 492 – 494), we specified that we referred to the water column: "This is also consistent with our previous study in twelve reservoirs (León-Palmero et al., 2023), and may indicate that denitrification is enhanced by particulate material derived from the phytoplankton community in the water column."*

Line 452 – 465: It is better to address the exact profiles shown in figures 2 & 3 when discussing oxycline turnover times and hypolimnetic storage, to help readers understand the statements.

*We thank the reviewer for the suggestion. We included the references to the figures in the revised manuscript. Please note that Figs. 2 and 3 became 3 and 4 in the revised manuscript (Lines 502 – 504): "The highest total N₂O production in Cubillas coincided with the highest N₂O concentration at the deepest depth in both months (Fig. 3). In the deeper reservoir, Iznájar, the*

*highest production was measured at the oxycline, where there is a strong potential for $N_2O$ fluxes, while the highest $N_2O$ concentrations were detected in the hypolimnion (Fig. 4)."*

**References:**

Bonin, P., Gilewicz, M. and Bertrand, J. C. (1989). Effects of oxygen on each step of denitrification on *Pseudomonas nautica*, *Canadian Journal of Microbiology*, 35(11), pp. 1061–1064. doi: 10.1139/m89-177.

Boström, K. H., Simu, K., Hagström, Å. and Riemann, L. (2004). Optimization of DNA extraction for quantitative marine bacterioplankton community analysis, *Limnology and Oceanography: Methods*, 2(11), pp. 365–373. doi: 10.4319/lom.2004.2.365.

Chen, H., Pan, H., Xiao, S. and Deng, S. (2025). Nitrous oxide dominates greenhouse gas emissions from hydropower's reservoirs in China from 2020 to 2060, *Water Research*, 279, p. 123420. doi: 10.1016/j.watres.2025.123420.

Granger, J. and Sigman, D. M. (2009). Removal of nitrite with sulfamic acid for nitrate N and O isotope analysis with the denitrifier method, *Rapid Communications in Mass Spectrometry*, 23(23), pp. 3753–3762. doi: 10.1002/rcm.4307.

Hochstein, L. I., Betlach, M. and Kritikos, G. (1984). The effect of oxygen on denitrification during steady-state growth of *Paracoccus halodenitrificans*, *Archives of Microbiology*, 137(1), pp. 74–78. doi: 10.1007/BF00425811.

Ion, I. V. and Ene, A. (2021). Evaluation of Greenhouse Gas Emissions from Reservoirs: A Review, *Sustainability*, 13(21), p. 11621. doi: 10.3390/su132111621.

Junier, P., Kim, O.-S., Witzel, K.-P., Imhoff, J. F. and Hadas, O. (2008). Habitat partitioning of denitrifying bacterial communities carrying *nirS* or *nirK* genes in the stratified water column of Lake Kinneret, Israel, *Aquatic Microbial Ecology*, 51(2), pp. 129–140. doi: 10.3354/ame01186.

Kim, O.-S., Imhoff, J. F., Witzel, K.-P. and Junier, P. (2011). Distribution of denitrifying bacterial communities in the stratified water column and sediment–water interface in two freshwater lakes and the Baltic Sea, *Aquatic Ecology*, 45(1), pp. 99–112. doi: 10.1007/s10452-010-9335-7.

León-Palmero, E., Morales-Baquero, R. and Reche, I. (2020). Greenhouse gas fluxes from reservoirs determined by watershed lithology, morphometry, and anthropogenic pressure, *Environmental Research Letters*, 15(4), p. 044012. doi: 10.1088/1748-9326/ab7467.

León-Palmero, E., Morales-Baquero, R. and Reche, I. (2023). P inputs determine denitrifier abundance explaining dissolved nitrous oxide in reservoirs, *Limnology and Oceanography*, 68(8), pp. 1734–1749. doi: 10.1002/lno.12381.

Lloyd, D., Boddy, L. and Davies, K. J. P. (1987). Persistence of bacterial denitrification capacity under aerobic conditions: The rule rather than the exception, *FEMS Microbiology Ecology*, 3(3), pp. 185–190. doi: 10.1111/j.1574-6968.1987.tb02354.x.

McIlvin, M. R. and Altabet, M. A. (2005). Chemical conversion of nitrate and nitrite to nitrous oxide for nitrogen and oxygen isotopic analysis in freshwater and seawater, *Analytical Chemistry*, 77(17), pp. 5589–5595. doi: 10.1021/ac050528s.

Pajares, S., Merino-Ibarra, M., Macek, M. and Alcocer, J. (2017). Vertical and seasonal distribution of picoplankton and functional nitrogen genes in a high-altitude warm-monomictic tropical lake, *Freshwater Biology*, 62(7), pp. 1180–1193. doi: 10.1111/fwb.12935.

Pjevac, P., Schauberger, C., Poghosyan, L., Herbold, C. W., van Kessel, M. A. H. J., Daebeler, A., Steinberger, M., Jetten, M. S. M., Lücker, S., Wagner, M. and Daims, H. (2017). *amoA*-targeted polymerase chain reaction primers for the specific detection and quantification of comammox *Nitrospira* in the Environment, *Frontiers in Microbiology*, 8. doi: 10.3389/fmicb.2017.01508.

Sigman, D. M., Casciotti, K. L., Andreani, M., Barford, C., Galanter, M. and Böhlke, J. K. (2001). A bacterial method for the nitrogen isotopic analysis of nitrate in seawater and freshwater, *Analytical Chemistry*, 73(17), pp. 4145–4153. doi: 10.1021/ac010088e.

Sturm, K., Yuan, Z., Gibbes, B., Werner, U. and Grinham, A. (2014). Methane and nitrous oxide sources and emissions in a subtropical freshwater reservoir, South East Queensland, Australia, *Biogeosciences*, 11(18), pp. 5245–5258. doi: 10.5194/bg-11-5245-2014.

Thamdrup, B. and Dalsgaard, T. (2002). Production of N2 through Anaerobic Ammonium Oxidation Coupled to Nitrate Reduction in Marine Sediments, *Applied and Environmental Microbiology*, 68(3), pp. 1312–1318. doi: 10.1128/AEM.68.3.1312-1318.2002.

Weigand, M. A., Foriel, J., Barnett, B., Oleynik, S. and Sigman, D. M. (2016). Updates to instrumentation and protocols for isotopic analysis of nitrate by the denitrifier method, *Rapid Communications in Mass Spectrometry*, 30(12), pp. 1365–1383. doi: 10.1002/rcm.7570.

Zumft, W. G. (1997). Cell biology and molecular basis of denitrification., *Microbiology and Molecular Biology Reviews*, 61(4), pp. 533–616.

---

## Author Comment (AC2)

**Please see below the point-by-point responses to Referee 2, and the actions taken regarding their concerns.**

**Major changes**

The revised manuscript includes relevant changes with respect to the submitted manuscript in order to address the reviewer's concerns. Please note that, due to the reorganization of certain sections, subsection numbering and figure references have been updated accordingly in the revised manuscript. In addition, we have replaced "predominant" with "dominant" in the title for grammatical correctness. In the text that follows, the suggestions and comments of the Referees are in black and plain font, and our responses are in *italics and blue font.*

Referee 2

**General comment:**

This study aims to investigate $N_2O$ production in two reservoirs and to distinguish its origin between nitrifying and denitrifying pathways. To achieve this, the authors combine natural-abundance isotopic analyses with rate measurements of $N_2O$ production associated with partial and complete nitrification as well as denitrification, together with molecular tools to quantify and trace the relevant metabolic pathways at the genetic level. The authors find that denitrification appears to be the main source of $N_2O$, with consistently higher $N_2O$ production rates and gene abundances than those associated with nitrification. The results highlight the value of combining isotopic and molecular approaches to understand nitrogen cycling in aquatic systems. The methodologies applied are well established. Overall, the manuscript addresses a timely and important topic in the context of climate change and contributes new insights into the origin of $N_2O$ production in lakes. The text is generally well written, although I suggest a minor reorganization of some sections to improve the flow (see specific comments below).

*We thank the reviewer for their thoughtful comment.*

**Specific Comments**

**Materials and Methods**

Reorganization of isotopic abundance section: I suggest moving the section on natural isotopic abundances so that it follows immediately after the "Vertical profiles and biogeochemical characterization" section and precedes the "Functional genes" section. Because isotopic abundances are part of the chemical characterization of the water column, presenting them earlier would improve the logical flow of the manuscript. If this restructuring is adopted, the corresponding results section should be reorganized accordingly, presenting the natural isotopic abundance results right after the physicochemical characterization and before the genetic characterization. While not essential, I believe this change would strengthen the overall structure.

*We thank the reviewer for their comments. We have reorganized the manuscript subsections to present the natural isotopic abundance earlier in the text. In the **Materials and Methods** section, the subsection **Natural abundance of stable isotopes (δ15N and δ18O)** is now subsection 2.3 (previously 2.9). It appears after **Vertical profiles and Biogeochemical characterization** and before **Functional genes.** Similarly, the subsection **Changes in concentration and isotopic composition of $N_2O$ and inorganic nitrogen** is now 3.2 (previously 3.5) and appears after the general description of vertical profiles. The*

*rearrangement required substantial rewording of several other sections, which we have done in order to be consistent and provide necessary context for the interpretation of the natural abundance data.*

*Following the reorganization of the figures, we considered it necessary to include the Chl-a profile in Figure 1, as it is discussed in the Results section (3.1).*

Subsection "Statistical tests": I recommend renaming this subsection to "Data analysis", which would allow the authors to describe more clearly the analytical criteria and tools used (e.g., the numbering system in the figures, the $\delta^{18}O$:$\delta^{15}N$ ratio, etc). Please also provide additional detail regarding the statistical procedures applied to linear and non-linear regressions. I assume that assumptions of normality, homoscedasticity, and independence were evaluated. Additionally, please specify the significance threshold used (e.g., $p < 0.05$).

*The subsection Statistical tests have been renamed as Data Analysis (2.10). We have included more details on the statistical procedures applied. Please see the revised text below:*

*"Statistical analyses were conducted in R (R Core Team, 2014) version 4.4.0. Data visualization was also performed in R, with final figure adjustments made using Inkscape (Inkscape Project, 2017). We assessed normality using the Shapiro-Wilk test of normality analysis and homogeneity of variances across groups using Levene's test. For normally distributed data with equal variances, we applied one-way ANOVA (F). When normality was met but variances were unequal, we used Welch's t-test; otherwise, the standard t-test was applied. For data that violated normality assumptions, we employed the Kruskal–Wallis rank-sum test (K–W) or the Wilcoxon test (W). Outliers were identified using the Grubbs test (G). Statistical significance was set at p < 0.05. Linear regressions were used throughout the study to evaluate the rates and drivers of $N_2O$ concentration and production. Model assumptions were assessed, and the model performance evaluated using adjusted $R^2$ values and predictor significance was determined using p-values (α = 0.05). Each sample was assigned a unique identifier (#1-12), which is shown in Table 1 and in the figures to facilitate data interpretation and highlight observed trends." Lines 229 – 238 in the revised manuscript.*

**Technical Comments**

Line 90: Please could you provide more detail on where and how the vertical profiles were measured? How many profiles were obtained per reservoir and sampling date? Was the same sampling site used in July and September?

*We have provided additional details in the revised version of the manuscript. Please see the new text (underlined) below:*

*"We sampled the water column near the dam, in the open water of the reservoir, at the same location during both the July and September campaigns. First, we conducted a vertical profile of the water column using a Sea-Bird 19plus CTD profiler, obtaining continuous measurements of temperature (ºC), dissolved oxygen (DO, µmol L$^{-1}$), and conductivity (µS cm$^{-1}$) in the reservoirs. Based on the temperature and DO profiles, we sampled three depths representing the epilimnion, oxycline, and hypolimnion or bottom waters. Water was collected at these three depths using a 5-L UWITEC bottle for further analyses and incubation experiments." Lines 95 – 100 in the revised manuscript.*

Line 118: Please specify which nosZ clade (I or II) was quantified.

*We have specified that we quantified clade I. Please see the text below:*

*"The nirS gene abundance was used as a proxy for denitrifiers, while nosZ gene (Clade I) abundance, was assessed only at the deepest layer, assayed only bacteria reducing $N_2O$ to $N_2$" Lines 156 and 157 in the revised manuscript.*

Line 124: Which was the headspace volume used for the oxic samples? Please, indicate it.

*The headspace during the oxic incubation was similar to the anoxic one (=3 mL). This detail has been included in the revised manuscript, as follows:*

*"Once in the lab, samples from oxic water depths (refer to Table 1) were purged uncapped for 2 min to remove excess $N_2O$, and a 3-mL headspace with ambient air was maintained after being exposed to ambient air for 30 min." Lines 161 – 163 in the revised manuscript.*

Line 212: A concentration >800 µmol $O_2$ $L^{-1}$ is unusually high (>25 mg $O_2$ $L^{-1}$) … Considering the DO profiles shown, it may be worth double-checking the calculation. For instance, if 16 mg O were used instead of 32 mg $O_2$ for the conversion, this could partly explain the discrepancy. I kindly suggest verifying this value to ensure consistency.

*We thank the reviewer for catching that error, the value should say 400 µmol $O_2$ $L^{-1}$. We have corrected the revised manuscript in the text (line 242) and in Figure 1.*

Figure 1a: All $N_2O$ concentration points for the Cubillas reservoir are the same color (orange). Additionally, the negative sign is missing from "-25" on the x-axis of the $^{15}N–NO_2^-$ panel.

*We thank the reviewer for the comment. Yes, all the points for Cubillas, also for Iznájar, are shown in orange for July, versus purple for September. We have clarified that in the figure caption as follows "The color scheme for all data is the same for both reservoirs: July (orange) and September (purple)."*

*We have added the missing sign to (-)25. We thank the reviewer for catching that error.*

Line 244: Please review spacing between symbols and values here and throughout the manuscript.

*We thank the reviewer for this observation. We have carefully reviewed the manuscript and corrected the spacing between symbols and values throughout to ensure consistency and proper formatting.*

Line 248: Please could you clarify more explicitly that these samples were excluded from the analysis?

*Please see the text below in the revised manuscript:*

*"These two samples, which were excluded from this analysis, contained the highest $NH_4^+$ concentrations (>6 µmol $L^{-1}$). The $N_2O$ production from $NH_4^+$ was an exponential function of the nitrification rates (Fig. 4b, adj $R^2$=0.60, value < 0.01)" Lines 295 – 297 in the revised manuscript.*

Line 256: Since there is no statistically significant relationship between the two variables, please reconsider the use of the word "coupled." "Accompanied by" would more accurately describe the pattern.

*The text has been modified following the reviewer's comment as follows:*

*"This decrease in the $NO_3^-$ reduction rates was accompanied by a decrease in the $NO_3^-$ concentration from July to September in both reservoirs" Lines 303 and 304 in the revised manuscript.*

Lines 257–258: The formula and interpretation of $NO_2^-$ turnover time (and $N_2O$ turnover time) would be more appropriately placed in the Methods section rather than in the Results. Including this information earlier would help readers better follow the analyses and their interpretation.

*We included this equation in the Methods section, in the subsection dedicated to $^{15}N$-$NO_2^-$ production (Lines 203 – 208 in the revised manuscript):*

*"Additionally, we also calculated the turnover time of $NO_2^-$ ($\tau_{NO_2^-}$, days), which represents the average time required to replace the nitrite pool given the measured production rate following equation (3):*

$$\tau_{NO_2^-} = \frac{[NO_2^-]}{R_{NO_2^-\ from\ NO_3^-}} \quad\quad\quad (3)$$

*where $[NO_2^-]$ represents the concentration of $NO_2^-$ (nmol-N $L^{-1}$), and $R_{NO_2^-\ from\ NO_3^-}$ represents the production rates of $NO_2^-$ from $NO_3^-$ (nmol-N $L^{-1}$ $d^{-1}$)".*

Line 261: Consider using p > 0.05 for non-significant results, and report exact p-values only when results are marginally significant. (Same comment for lines 269, 270, and 275.)

*We thank the reviewer for this suggestion. We have revised the manuscript to follow this recommendation: non-significant results are now reported as p > 0.05, and exact p-values are provided only for results that are marginally significant. This change has been applied consistently to those lines.*

Figures 2 and 3: What do the numbers displayed next to $N_2O$ concentrations represent? Please clarify this in the figure caption. The colour coding is also confusing: orange is used both for July samples and for $N_2O$ production, regardless of sampling date. Please consider selecting a different colour for $N_2O$ production.

*The numbers displayed next to $N_2O$ concentrations correspond to the sample IDs, which are initially reported in Table 1. We have clarified this in the figure captions for Figures 2 and 3 (3 and 4 in the revised manuscript).*

*Additionally, we have revised the colour scheme to avoid confusion in these figures. In the revised figures, the $N_2O$ production is shown in magenta instead of orange.*

Line 290: The dark-gray sediment colour referenced in the caption is not visible in any panel of Figure 3. Please remove this part of the caption.

*We have removed that part of the caption.*

Figure 4: Please consider using a lighter colour (or open symbols) for the excluded data points. As currently displayed, they are somewhat difficult to distinguish.

*We thank the reviewer for this suggestion. To improve clarity, we have used open symbols for the excluded data points, which contrasts with the colors used for the included data.*

Figure 5: Please explain what the numbers represent, ideally in the caption. Additionally, clarifying in the Methods how these numbered points relate to those in Figures 2 and 3 would help guide the reader through the Results and Discussion.

*The numbering scheme for the samples has now been introduced in the Methods, which should clarify the figures. Please see the following text in the subsection "2.10. Data Analysis": "Each sample was assigned a unique identifier (#1-12), which is shown in Table 1 and in the figures to facilitate data interpretation and highlight observed trends." Lines 237 and 238 in the revised manuscript.*

*Additionally, we included the explanation in the caption as follows: "Correspondence between numbers and samples is shown in Table 1 and Figs. 2 and 3".*

Figure 5a: Why is the segment connecting points 11 and 12 shown in red? I could not find an explanation in the text.

*We thank the reviewer for noticing this. The red segment connecting points 11 and 12 in Figure 5a indicates a trend associated with $N_2O$ consumption, as opposed to the black segments that represent trends associated with $N_2O$ production. We have now clarified this in the figure caption, and in the text to avoid confusion. Besides, we added an extra legend in panel (a).*

Figure 5c: There appear to be two red dotted lines. Which one is valid? Could you please specify and clarify this in the caption? I additionally suggest explaining the use of this ratio more clearly in the "Data analysis" section.

*Both dotted lines are correct, and represent the ratio $\delta^{18}O:\delta^{15}N$ (2.5). To avoid confusion with the red lines in (a), we changed the color to green. The use of this ratio, which is indicative of active $N_2O$ reduction (Ostrom et al., 2007), is explained in the discussion.*

Line 314–315: Please could you provide a reference for the threshold defining suboxic conditions (DO < 10 µmol L$^{-1}$).

*The threshold of 10 µM $O_2$ was chosen following the operational definition provided in the Springer Nature Encyclopedia of Astrobiology entry "Suboxic," which states that the boundary between hypoxic and suboxic conditions is widely taken as 10 µM $O_2$. We have also used this threshold in previous works (e.g., León-Palmero et al., 2023).*

*The references have been provided in the main text as follows:*

*"In both reservoirs, the higher $N_2O$ concentrations were found in the deepest layers under suboxic conditions (i.e., DO < 10 µmol L$^{-1}$) (León-Palmero et al., 2023; Pinti, 2014)" Lines 345 and 246 in the revised manuscript*

Line 317: The manuscript uses the term "relationship" for *nirS* vs. DO concentration, but "correlation" for nirS vs. cumulative Chl-a. Please use consistent terminology throughout.

*We thank the reviewer for this comment. We have revised the manuscript to ensure consistency and replaced "relationship" with "correlation" where appropriate. For example, see lines 349 - 350 in the revised manuscript:*

*"It is thus consistent that nirS abundance showed a negative underline{correlation} with DO concentration (Fig. 6d) and a positive underline{correlation} with cumulative Chl a concentration (Fig. 6e)."*

Line 367–368: Given the lack of detection issues for AOA in the Cubillas reservoir in September, I am not fully convinced that the presence of AOB and Comammox can be dismissed as easily.

> *Thank you for your comment. We agree that AOB and Comammox cannot be entirely ruled out. Our interpretation was based on previous evidence showing AOA dominance in these reservoirs (León-Palmero et al., 2023) and the absence of bacterial amoA measurements in this study. However, we acknowledge that pre-filtration and the lack of targeted analysis for AOB and Comammox may have limited our ability to detect these groups. We have revised the manuscript to clarify this point and avoid overgeneralization. Please see the revised text below:*

> *"Previous work in San Francisco Bay revealed that dominant AOA clades were not amplified by commonly used primers, including those employed in this study (Rasmussen and Francis, 2022). It is therefore possible that important AOA lineages present in these reservoirs were missed, leading to an underestimation of amoA abundance. We did not measure the bacterial amoA gene abundance, because AOA had previously been identified as the dominant ammonia-oxidizers in the study reservoirs (León-Palmero et al., 2023). Therefore, we cannot assess the potential contribution of AOB. We tested for Comammox using specific primers and did not detect them in any sample. Additionally, sample water was pre-filtered before DNA extraction (pore size = 3 µm), which may have excluded microbes attached to particles or suspended sediment, potentially including AOA or Comammox groups." Lines 407 – 415 in the revised manuscript.*

Line 375–376: Because no positive control for Comammox was available, the absence of amplification does not allow the rejection of the hypothesis that high nitrification rates without ammonium oxidation could be due to complete ammonia oxidation. I encourage the authors to consider this possibility.

> *We agree that the absence of amplification without a positive control does not allow us to conclusively reject the presence of Comammox. We have revised the text to acknowledge this limitation and to consider the possibility that complete ammonia oxidation could explain high nitrification rates without detectable ammonium oxidation intermediates. Please see the revised text below (Lines 418 – 423) :*

> *"The detection of high nitrification rates, but no significant ammonia oxidation, might suggest that comammox is occurring at these depths. However, our PCR analysis showed no evidence of the presence of comammox bacteria (Fig. S2), although, because no positive control was available, we cannot completely exclude their presence. Therefore, we consider the possibility that complete ammonia oxidation could contribute to the observed nitrification rates. Alternatively, we hypothesize that the $NO_2^-$ production by ammonia oxidation was tightly coupled to $NO_2^-$ consumption by $NO_2^-$ oxidizers, such that it could not be detected in the $NO_2^-$ pool."*

Line 411–412: If the earlier suggestion is incorporated, this description should be moved to the "Data analysis" section.

> *Thank you for your suggestion. We have incorporated the description into the "Data Analysis" subsection as recommended. However, we have also kept a brief mention in the Discussion to maintain clarity for readers who may not refer back to the methods while interpreting the results. We believe this helps contextualize the interpretation without redundancy.*

Line 415: To support the interpretation, "which indicates net $N_2O$ production" should specify by which process (i.e., denitrification, AOB, or Comammox).

> *We modified the text as follows (Lines 462 – 464):*
>
> *"In general, the increase in the $N_2O$ concentration with depth was coupled to the $\delta^{15}N$-$N_2O$ decrease (e.g., #1-3, #5-6 or #7-9 in Figs. 1 and black trend lines in 2a), which indicates net production of $N_2O$ by water column denitrification, nitrifier denitrification and/or bacterial nitrification"*

Lines 417–418: The term "coupling" implies a relationship between variables that is not statistically supported. Please consider using "accompanied by" instead.

> *We have applied the reviewer's suggestion and replaced "coupling" with "accompanied by" in the text. The revised sentence now reads:*
>
> *"There was also an increase in the $\delta^{18}O$-$N_2O$ with depth in each profile, accompanied by an increase in $N_2O$ concentration, which also suggests a parallel production and consumption of $N_2O$ at the deeper layers" Lines 466 – 468 in the revised manuscript.*

Lines 432–440: These results are very interesting, and the discussion provided here is excellent!

> *We thank the reviewer for the comment.*

**References:**

Inkscape Project: Inkscape: Open Source Scalable Vector Graphics Editor, 2017.

León-Palmero, E., Morales-Baquero, R., and Reche, I.: P inputs determine denitrifier abundance explaining dissolved nitrous oxide in reservoirs, Limnology and Oceanography, 68, 1734–1749, https://doi.org/10.1002/lno.12381, 2023.

Ostrom, N. E., Pitt, A., Sutka, R., Ostrom, P. H., Grandy, A. S., Huizinga, K. M., and Robertson, G. P.: Isotopologue effects during N2O reduction in soils and in pure cultures of denitrifiers, Journal of Geophysical Research: Biogeosciences, 112, https://doi.org/10.1029/2006JG000287, 2007.

Pinti, D. L.: Suboxic, in: Encyclopedia of Astrobiology, edited by: Gargaud, M., Amils, R., Quintanilla, J. C., Cleaves, H. J., Irvine, W. M., Pinti, D. L., and Viso, J. V., Springer, Berlin, Heidelberg, https://doi.org/10.1007/978-3-642-27833-4_1463-2, 2014.

R Core Team: R: A Language and Environment for Statistical Computing, R Foundation for Statistical Computing, Vienna, Austria, 2014.

Rasmussen, A. N. and Francis, C. A.: Genome-Resolved Metagenomic Insights into Massive Seasonal Ammonia-Oxidizing Archaea Blooms in San Francisco Bay, mSystems, 7, e01270-21, https://doi.org/10.1128/msystems.01270-21, 2022.